



## Can label or protein deuteration extend the phase relaxation time of Gd(III) spin labels?

Elena Edinach[1][#], Xing Zhang[2][#], Chao-Yu Cui[2], Yin Yang[2], George Mitrikas[3], Alexey Bogdanov[1], Xun-Cheng Su[2,*], and Daniella Goldfarb[1,*]

[1]Department of Chemical and Biological Physics, Weizmann Institute of Science,76100 Rehovot, Israel

[2]State Key Laboratory of Elemento-Organic Chemistry, College of Chemistry, Nankai University, 300071 Tianjin, China

[3]Institute of Nanoscience and Nanotechnology, NCSR Demokritos, Athens 15310, Greece

Corresponding authors : Xun-Cheng Su, xunchengsu@nankai.edu.cn, Daniella Goldfarb, Daniella.goldfarb@weizmann.ac.il

[#] Equal contributions



## 1 Abstract

Pulse-dipolar electron paramagnetic resonance (PD-EPR) has emerged as an effective tool in
structural biology, enabling distance measurements between spin labels attached to
biomolecules. The sensitivity and the accessible distance range of these measurements are
governed by the phase memory time ($T_m$) of the spin labels. Understanding the decoherence
mechanisms affecting $T_m$ is crucial for optimizing sample preparation and spin-label design. This
study investigates the phase relaxation behavior of two Gd(III) spin-label complexes, Gd-PyMTA
and Gd-TPMTA, with various degrees of deuteration. These two complexes have significantly
different zero-field splitting (ZFS) parameters. Hahn echo decay and dynamical decoupling (DD)
measurements were performed at W-band (95 GHz) in deuterated solvents ($D_2O$/glycerol-$d_8$),
both for the free complexes and when conjugated to proteins. The impact of temperature,
concentration, and field position within the EPR spectrum on $T_m$ was examined. Results indicate
that protons within 5 Å of the Gd(III) ion do not contribute to nuclear spin diffusion (NSD), and
protein deuteration offers minimal enhancement in $T_m$. The dominant phase relaxation
mechanisms identified at low concentrations were direct spin-lattice relaxation ($T_1$) and
transient ZFS fluctuations (tZFS). Dynamical decoupling (DD) measurements, using the Carr-
Purcel sequence with ~140 refocusing pulses, resolved the presence of two populations: one
with a long phase relaxation time, $T_{m,s}$, and the other with a short one, $T_{m,f}$. The dominating
mechanism for the slowly relaxing population is direct-$T_1$. $T_{m,s}$ showed no concentration
dependence and was longer by a factor of about 2 from $T_m$ for both complexes. We tentatively
assign the increase in $T_{m,s}$ to full suppression of the residual indirect $T_1$-induced NSD mechanism.
For the fast relaxing population, $T_{m,f}$ is shorter for Gd-TPMTA; therefore, we assign it to
populations for which the tZFS mechanism dominates. Because of the relatively short $T_1$ and the
contribution of tZFS mechanism, protein deuteration does not significantly affect $T_m$.



## 1. Introduction

Distance measurements between two spin labels attached at specific sites in biomolecules, determined by pulse-dipolar EPR (PD-EPR) methods, have become standard tools in structural biology. The sensitivity of these measurements and the distance they can access depend on the phase memory time, $T_m$, of the spin labels used. Accordingly, understanding the decoherence mechanisms is essential for optimizing sample preparation conditions (concentrations, solvent composition, deuteration) and for spin-label design, thus reaching as long as possible $T_m$. In the case of solid-state EPR where the pulses applied cannot excite the entire width of the EPR spectrum, the spins are commonly divided into two types: Those excited and observed by the microwave pulse are referred to as the "A" spins, and the rest, termed "B" spins, are much more abundant. In general, at low temperatures, at which PD-EPR experiments are commonly carried out, there is no motion and the mechanisms contributing to decoherence for spin labels with S=1/2 are (Salikhov et al., 1981; Tyryshkin et al., 2012; Mitrikas, 2023; Wilson et al., 2023; Eaton and Eaton, 2000): (i) Direct spin-lattice ($T_1$) relaxation mechanism of the A spins, $T_{m,T1}$, which provides the highest limit $T_m \leq 2T_1$ .(ii) Redistribution of resonance frequencies of dipole-coupled A spins due to mw pulses leads to instantaneous diffusion, ID. (iii) Coupling between A spins and nearby B spins leads to spectral diffusion, SD. The latter can result from $T_1$ flips of the coupled B spins, referred to as indirect-$T_1$, SD-$T_1$, or energy-conserving pairwise B spin flip-flops, SD-ee. All these are concentration-dependent. (iv) Nuclear spin diffusion (NSD) arising from nuclear flip-flops caused by homonuclear couplings, which does not depend on the electron spin concentration but depends on the nuclei concentration. (v) Admixture of tunnel states of methyl groups into the electron spin mediated by the hyperfine coupling of methyl protons (Soetbeer et al., 2021a).

Gd(III) chelates are among the spin labels used for PD-EPR applications; they are beneficial for in-cell PD-EPR measurements because of their chemical stability and high sensitivity at high frequencies (> 34 GHz, Q- and W-band) owing to a narrow central transition and a relatively long phase memory time (Goldfarb, 2014; Giannoulis et al., 2021). Several studies have been dedicated to the dephasing mechanism of Gd(III) at low temperatures. Raitsimring et al (Raitsimring et al., 2014) explored the phase relaxation of Gd(III)-DOTA as a representative of Gd(III) spin labels in the temperature and concentration ranges typically used for W-band double electron-electron (DEER) measurements, which is the most widely applied PD-EPR

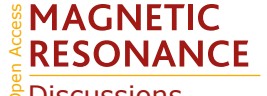

experiment.(Pannier et al., 2011) They found that in addition to the mechanisms of phase
relaxation known for nitroxide-based spin labels listed above, Gd(III) spin labels are subjected to
an additional phase relaxation mechanism that features an increase in the relaxation rate from
the center to the periphery of the EPR spectrum. It was suggested that this mechanism is due to
transient zero-field splitting (tZFS) fluctuations. This tZFS-induced phase relaxation mechanism
becomes dominant (or at least significant) when all other phase relaxation mechanisms
mentioned above are significantly suppressed by matrix (solvent) deuteration and low spin
concentration.
A quantitative analysis of Gd(III) Hahn echo decay was recently reported at 240 GHz (Wilson et
al., 2023). Two complexes, Gd-DOTA ($D$=700 MHz) and Gd-PyMTA ($D$=1200 MHz), were studied
in $D_2O$/glycerol-$d_8$. $T_1$ and $T_m$ were measured as a function of temperature and concentration. As
expected $T_1$ was found to be concentration-independent, whereas $T_m$ was. Interestingly, the
Hahn echo decay from which $T_m$ was derived could be fitted by an exponential decay instead of
the stretched exponent needed at Q-band (Soetbeer et al., 2021b). A careful analysis of the
temperature and concentration dependence data and the associated $T_1$ values gave the relative
contributions of the various decoherence mechanisms. The concentration-independent
mechanism was found to be the direct-$T_1$ mechanism and a concentration- and temperature-
independent mechanism was assigned to weak coupling between electron spins and the
presence of an ensemble of nuclear spins. As the solvent was fully deuterated, these could be
protons on the Gd(III) complex. In this respect, it has been shown that while deuteration of
nitroxides spin labels did not increase $T_m$, for trityl labels, it did (Soetbeer et al., 2021b).
Qualitative studies of several Gd(III) complexes, with axial parameters, $D$, of the ZFS in the range
of 560-2000 MHz, investigated the effect of solvent deuteration at Q-band(Garbuio et al., 2015),
and reported that in a protonated matrix, $T_m$ is dominated by NSD. It was also suggested that in
fully deuterated solvents, the phase relaxation is dominated by tZFS and perhaps ligand
hyperfine-driven mechanism, but no quantitative analysis of the data was presented. The
decoherence behavior of Gd(III) complexes was further investigated under dynamical
decoupling (DD) conditions. DD is a control strategy to protect quantum states from
decoherence, achieved by applying a sequence of carefully designed control pulses that
counteract unwanted interactions with the environment, effectively "decoupling" the system
from environmental noise.(Suter and Álvarez, 2016) CP (Carr-Purcell) and CPMG (Carr-Purcell-



Meiboom-Gill) echo trains are such sequences. DD acts as a filter between the spin system and
the environment, and the pulse spacing determines the characteristics of this filter. Short delays
compared to the correlation time of the environmental fluctuations increase the coherence time
of the system and typically, relaxation caused by electron-electron spectral diffusion and NSD
can be suppressed (Soetbeer et al., 2021b; Soetbeer et al., 2018). Recent measurements on a
single crystal of Gd(III) doped Y(trensal) carried out at X-band frequencies showed that CPMG
with 120 refocusing pulses suppressed NSD efficiently  and increased $T_m$ considerably,
depending on the transition probed and the crystal orientation (Hansen et al., 2024).
Three Gd(III) complexes with $D$ values of 485-1861 MHz were studied at Q-band in protonated
and deuterated solvent, $H_2O$/glycerol and $D_2O$/glycerol-$d_8$, and a low concentration to suppress
the ID and SD mechanisms (Soetbeer et al., 2021b). As expected, the solvent deuteration
increased the decay time considerably, and the data could be well-fitted with a single stretched
exponential decay function (SE model). Furthermore, for Gd-DOTA-M in deuterated solvents, DD
(CP with up to 5 refocusing pulses) did not generate a significant increase in the decay time, and
it was suggested that coherence losses of unknown origin, probably the ZFS-driven mechanism,
which the DD cannot refocus, counteracted the decoupling efforts.(Soetbeer et al., 2021b)
Interestingly, measurements of a protein singly spin-labeled with Gd-DOTA-M in a deuterated
solvent increased the echo decay rate approximately threefold, as compared to the bare spin-
label in the same solvent. This three-fold increase could be counter-acted by DD with 2-3 pulses,
achieving overall longer coherence survival than any DD trace of the fully protonated sample
(Soetbeer et al., 2021b). This showed that the protein's protons do affect phase relaxation. Thus,
one would expect that deuteration of the protein should help reduce the decoherence.
In the present work, we explore whether deuteration of the Gd(III) chelate and the protein can
further extend $T_m$ , along with the effects of concentrations, temperature, and field position
within the Gd(III) EPR spectrum. This is done for Hahn echo decay measurements, and the
potential of DD for extending it further is also examined. We studied two Gd(III) spin-label
complexes, PyMTA (Gd-PyMTA) and TPMTA (Gd-TPMTA) (**Fig. 1**), with different degrees of
deuteration. These two complexes have very different $D$ values (1200 vs ~4000 MHz). All
measurements were carried out at W-band (95 GHz), and the complexes were dissolved in
$D_2O$/glycerol-$d_8$, thus serving as a reference for the longest possible $T_m$. These are then
compared to the phase relaxation of these labels when conjugated to a protein. We found that





the protons at a distance shorter than 5 Å from the Gd(III) do not contribute to NSD and that the
protein's deuteration did not significantly prolong $T_m$. The primary mechanisms contributing to
the phase relaxation in deuterated solvents and low concentrations are the direct $T_1$ and tZFS
mechanisms.

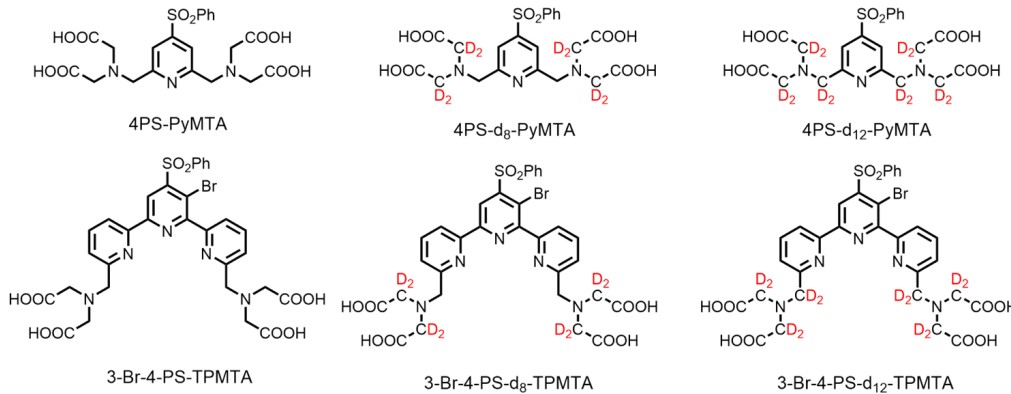

*Figure 1. Deuterated chelates for Gd(III) that can be further attached to cysteine residues in a*
*protein for labeling.*
**1. Experimental**
1.1 Synthesis of spin labels
The synthesis of spin tags, protein expression and labeling are described in detail in the
Supporting Information. In brief, the 4PS-($d_n$)-PyMTA were synthesized according to reported
procedure (Montgomery et al., 2017; Wang et al., 2019) (Li and Byrd, 2022), (Yang et al., 2015))
and 3-Br-4-PS-($d_n$)-TPMTA tags were synthesized in the following 8 steps (Scheme 1). 1:
Methylation of (3-1) with $CH_3I$ and $K_2CO_3$ in acetone, purification via column chromatography
(82% yield). 2: Bromination of (3-2) with N-Bromusuccinimide (NBS) in acetic acid at 60 °C, and
chromatography purification (92% yield). 3: Reduction of (3-3) with $NaBH_4/NaBD_4$ in ethanol,
yielding a yellow solid. 4: Reaction of (3-4) with $PBr_3$ in $CHCl_3$, followed by neutralization and
extraction. 5: Substitution of (3-5) with diethyl iminodiacetate in acetonitrile at 70 °C, yielding a
yellow solid. 6: Reaction of (3-6) with $POBr_3$ in DMF at 105 °C and purification. 7: Reaction of (3-
7) with sodium benzene sulfinate and TBAB in acetonitrile at 90 °C. 8: Hydrolysis of (3-8) with
$NaOH/NaOD$ in $THF/H_2O$, followed by acidification, resulting in (3-9) as a yellow solid. The
synthesis of 4PS-5-Br-6PCA-($d_n$)-DO3A-Gd(III) is described in detail in the Supporting
Information.



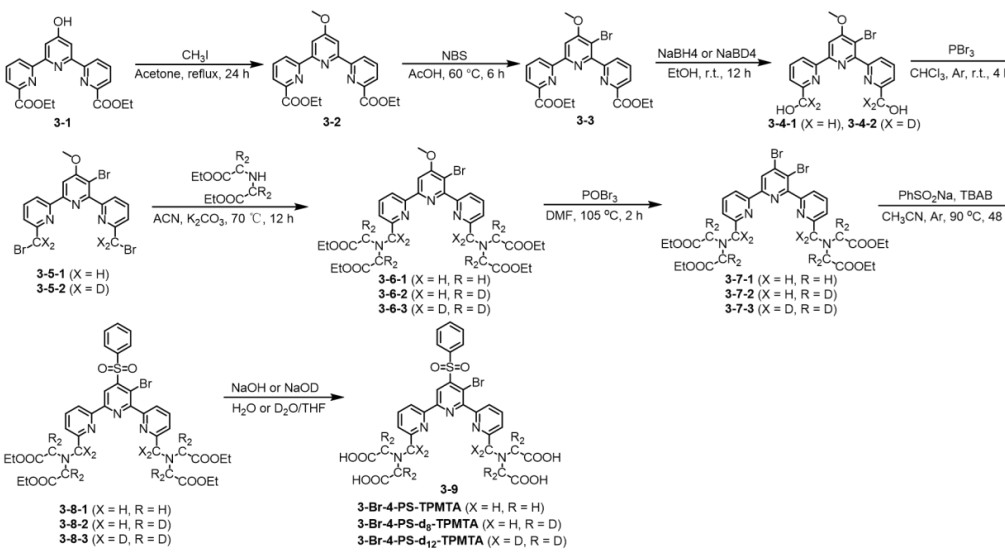

***Scheme 1.*** *Synthesis of 3-Br-4-PS-($d_n$)-TPMTA spin tags.*

2.2 Protein purification, spin labeling, and EPR sample preparation
The D39C/E64C construct of ubiquitin (ub) was used in this study. The steps for deuterated
protein expression were performed according to previous reports (Li and Byrd, 2022).
*2.2.1    Protein labeling with 4PS-5-Br-6PCA-($d_n$)-DO3A-Gd(III).*
0.2 mM 100 µL purified protein ($^1$H $^{14}$N ubi D39C/E64C or $^2$H $^{15}$N ub D39C/E64C) was incubated
with 0.4 mM tris(2-carboxyethyl)phosphine (TCEP) in 20 mM Tris-HCl at pH 8.5, and then treated
with 10 equivalents of tags at 30 °C for 12 h. The reaction progress was monitored by ESI-Q-TOF
mass spectrometry. The excess tag was removed using a PD-10 desalting column (GE Healthcare
Biosciences). The ligation products were freeze-dried for subsequent experiments.
*2.2.3    Protein labeling with 4PS-PyMTA or 3-Br-4PS-TPMTA.*
The ligation of the target protein to 4PS-PyMTA was carried out according to the previous
reports (Yang et al., 2019). 0.2 mM 100 µL purified protein in 20 mM Tris-HCl at pH 8.5 was
mixed with 0.4 mM TCEP, and then treated with 10 equivalents tags at 30 °C for 12 h. After the
reaction was completed, the sample was filtered through PD-10 desalting column to remove the
excess tag. The protein-PyMTA was mixed with 2.5 equivalents of Gd(NO$_3$)$_3$ in a 20 mM MES
buffer at pH 6.5. The excess of metal ion was removed using a Millipore concentrator (3 kDa
cutoff). Similarly, 3-Br-4PS-TPMTA was conjugated to the target protein using the same
procedure, and the deuterated tags were ligated to the target protein the same as the non-
deuterated tag.
For pulse EPR measurements, Gd-PyMTA and Gd-TPMTA solutions in the concentration range of
0.03-0.2 mM were dissolved in 50:50 v/v $D_2O$/glycerol-$d_8$. The spin-labeled protein conjugates
were lyophilized and redissolved in 15 mM HEPES-$D_2O$ buffer (pD 7.2) with 20% glycerol-$d_8$ (v:v).
The final concentration of proteins was 50 µM estimated from the absorbance at 280 nm using a
Nanodrop spectrophotometer (Thermo Science). For EPR measurements, solutions (ca. 3 µL)
were transferred to quartz capillaries (0.6 ID × 0.84 OD mm) and sealed at one end with
crytoseal.
*2.3 Spectroscopic measurements*
Pulsed EPR and ENDOR measurements were performed using two home-built W-band pulse EPR
spectrometers equipped with cylindrical $TE_{011}$ cavities and Helmholtz radiofrequency (RF) coils
(Gromov et al., 1999). The first spectrometer has a solenoid superconducting magnet
(Cryomagnetics, Inc.), a 3 W pulsed microwave power amplifier (QPP95013530, Quinstar), and a
pulsed 2 kW RF amplifier (BT02000-GammaS, TOMCO). The second spectrometer has a 0–5 T
cryogen-free magnet with an integrated variable temperature unit and 300 mT sweep coil
(J3678, Cryogenic Ltd.)(Feintuch et al., 2011), and is equipped with 2 W pulsed microwave
power amplifier (QPP95023330-ZW1, Quinstar). All temperature and field dependencies of Gd -
TPMTA- were carried out using the second spectrometer due to its wide temperature and
magnetic field ranges.
Echo-detected EPR (ED-EPR) spectra were recorded employing the Hahn echo sequence ($\pi/2-\tau-$
$\pi-\tau-echo$) sequence and measuring the echo intensity as a function of the magnetic field. The $\pi$
pulse duration was 28-30 ns, $\tau$=500-600 ns, and a repetition time of 1 ms. Echo decays as a
function of $\tau$ were measured by setting the magnetic field to the maximum of the ED-EPR
spectra with the experimental parameters described above. The Carr–Purcell (CP) based scheme
experiments were carried out using the $\pi/2 - (\tau/n - \pi - \tau/n)_n - echo$ sequence with a $2^n$-step
phase cycling employed to filter out all additional echoes except the refocused ones (Soetbeer
et al., 2018). For these experiments, n was from 1 to 5 with varying $\tau$ for each value of n and
measuring the intensity of the last echo as a function of $\tau$. Additionally, a full CP train $\pi/2_x - (\tau -$
$\pi_x - \tau - echo - \tau - \pi_{-x} - \tau - echo)_n$ (Mentink-Vigier et al., 2013) was applied with a two-step phase
cycling on the first $\pi/2$ pulse, with constant $\tau$ in the range of 280 to 800 ns and the intensity of



each echo was measured, typically n~140. $T_1$ measurements for were performed using the
inversion recovery sequence, $\pi - t_{wait} - \pi/2 - \tau - \pi - \tau -$ echo, with varying $t_{wait}$.
Mims ENDOR spectra were recorded on the first spectrometer at 10-11K and a magnetic field
corresponding to maximum echo intensity using the sequence $\pi/2-\tau-\pi/2-T(\pi_{RF})-\pi/2-\tau-$echo$-$
$[\tau_2-\pi-\tau_2-$echo$]_n$ with a four-step phase cycle and five CP echoes with $\tau_2$ = 600 ns for detection,
which was optimized for the best signal-to-noise ratio (Mentink-Vigier et al., 2013) The RF
frequency was varied randomly (Epel et al., 2003) The experimental parameters for the Mims
ENDOR spectra were T=42 µs, τ varied from 280 ns to 600 ns. RF power was adjusted to yield the
desired $\pi_{RF}$ pulse length (40 µs), using a Rabi nutation sequence, $\pi/2-\tau-\pi/2-T(t_{RF})-\pi/2-\tau-$echo,
with a constant mixing time, T, of 100 µs and varying RF pulse length, $t_{RF}$.
2.4. Simulations of the ED-EPR spectra:
ED-EPR spectra were simulated using EasySpin program package (Stoll and Schweiger, 2006)
using solid-state simulation function "pepper". The distributions of ZFS parameters were
considered using a built-in EasySpin functionality (DStrain parameter), and the Boltzmann
thermal polarization of the electron spin levels at the W-band at the temperature of the
experiment was taken into account. A Gaussian line shape with 0.1 mT width was used for
simulations. To account for the difference in turning angles for different electron spin manifolds
in a pulsed ED-EPR experiment, intensities of individual transitions $\left|m_S\right\rangle \leftrightarrow \left|m_S+1\right\rangle$ were
renormalized according to $\sin^3\left(\pi\alpha/2\right)\cdot\alpha^{-1}$ (Raitsimring et al., 2013) where
$\alpha = \sqrt{S(S+1)-m_S(m_S+1)}\Big/\sqrt{S(S+1)+0.25}$ and $S=7/2$. This approach still does not
consider the difference in phase memory times of different electron spin manifolds, which is
minor for the short inter-pulse τ delays used in the ED-EPR sequence (500–600 ns ). The optimal
values of the parameters were determined by non-linear least-squares fitting.
**3. Results and discussion**
3.1 Deuterated Gd(III) spin labels
*3.1.1 ED-EPR and ENDOR spectra*
Before proceeding with the relaxation measurements, we carried out spectroscopic
characterization of the samples. The W-band echo detected EPR (ED-EPR) spectra of Gd-PyMTA
and Gd-TPMTA, recorded at 10K, are shown in **Fig. 2A,B**. The spectrum of Gd-PyMTA is typical
for Gd(III) spin labels with a moderate ZFS (*D*=1200 MHz)[27] in frozen solutions where the $m_S=-$



1/2 to $m_S$=1/2 central transition (CT) dominates and appears as an intense peak superimposed
on a broad featureless background arising from all other transitions. The unresolved broad
background results from a large distribution in the ZFS parameters $D$ and $E$.(Raitsimring et al.,
2005) The spectrum of Gd-TPMTA is unusual; the central transition has a fine structure, and the
broad background on which it is superimposed has clear singularities. This indicates that the ZFS
is considerably larger than for Gd-PyMTA and that the distributions of $D$ and $E$ are smaller. To
ease the assignment of the various features of the spectrum, we recorded the spectrum at
lower temperatures, where the contribution of the CT decreases and those of the low-lying
transitions $|-7/2>\rightarrow|-5/2>$ and $|-5/2>\rightarrow|-3/2>$ increase. The spectra are presented in **Fig. 2C**
with the annotation of the powder pattern's x, y, and z singularities corresponding to the various
transitions. Simulations of the spectra presented in **Fig. 2D** gave $D$=4200 MHz, $\Delta D$=390 MHz,
$E$=440 MHz, and $\Delta E$=370 MHz. We attribute the larger ZFS values and the smaller distributions
to TPMTA, offering optimal 9 coordination sites for Gd(III), holding it in a well-defined position
as opposed to PyMTA, which has 7 coordination sites, and the other two are supplemented by
water molecules.

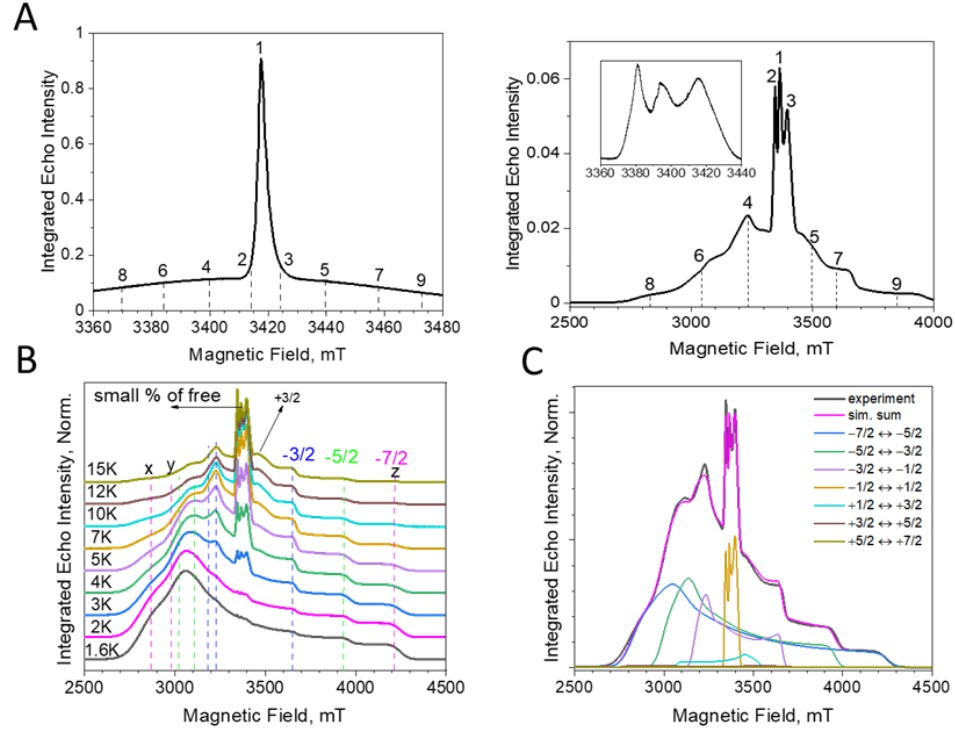

*Figure 2. ED-EPR spectra (10 K) of Gd-PyMTA (A) and Gd-TPMTA (B). The central transition of Gd-TPMTA is shown in the insert on the right with an extended scale. The different numbers indicate positions at which relaxation measurements took place. C) Temperature-dependent ED-EPR spectra of Gd-TPMTA. The positions of the x,y,z singularities of the powder patterns of the various transitions are indicated. D) Simulations of the spectrum in (C) recorded at 5K, the simulation parameters are given in the text.*

Next, we carried out W-band Mims ENDOR measurements to test the efficiency of the deuteration and determine the hyperfine couplings of the different protons, which are important for identifying their potential contributions to decoherence by NSD. **Fig. 3A,B** presents the spectra of Gd-PyMTA, Gd-PyMTA-$d_8$, and Gd-PyMTA-$d_{12}$ measured with $\tau$=280 ns to highlight the large $^1$H couplings and $\tau$=600 ns to highlight the small $^1$H couplings.

For Gd-PyMTA-$d_{12}$ the spectrum is dominated by the protons on the pyridine ring, having a coupling $a_\perp$=440 kHz, and the protons on the phenyl rings with $a_\perp$=140-170 kHz, where $a_\perp$ is the principal, perpendicular component of hyperfine tensor. A comparison of the spectra of the three Gd-PyMTA samples shows some residual methylene protons. The ENDOR spectra of Gd-TPMTA, Gd-TPMTA-$d_8$, and Gd-TPMTA-$d_{12}$ are presented in **Fig. 4C,D**. In this case, the remaining



protons of Gd-TPMTA-$d_{12}$ are situated on three pyridine rings and are located 5.5 and 6.5-7 Å
away from the Gd(III) with $a_\perp$=170-440 kHz. Here, the deuteration efficiency was higher than
that of Gd-PyMTA-$d_{12}$, as we did not observe a significant contribution of residual methylene
protons. A summary of the hyperfine couplings of the various protons in Gd-PyMTA and Gd-
TPMTA is given in **Table S1**.

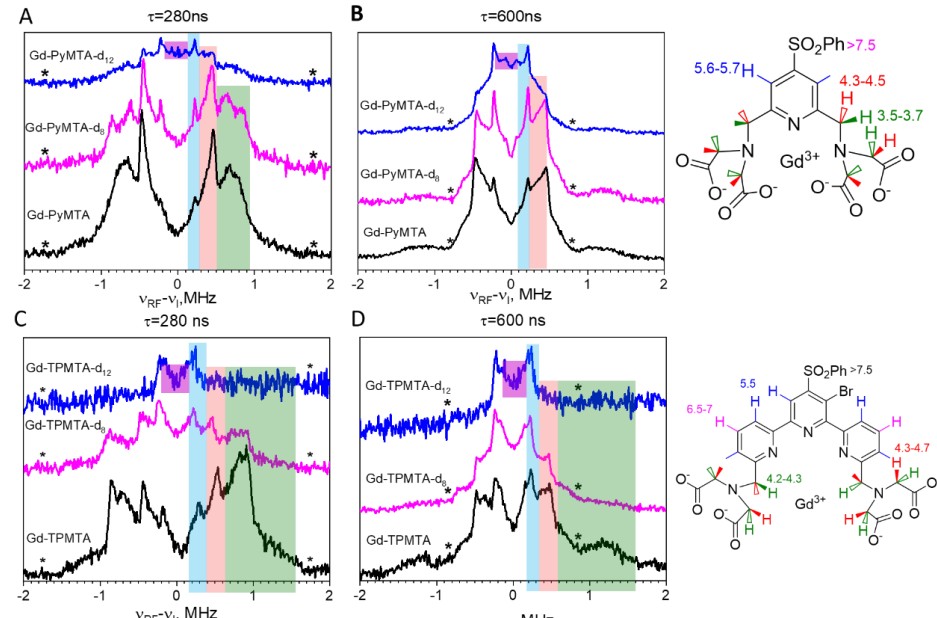

***Figure 3. (A,B)****Mims ENDOR spectra of Gd-PyMTA, Gd-PyMTA-$d_8$ and Gd-PyMTA-$d_{12}$ measured at*
*the CT with two $\tau$ values (indicated on the figure). (C,D) Mims ENDOR spectra of Gd-TPMTA, Gd-*
*TPMTA-$d_8$, and Gd-TPMTA-$d_{12}$, measured at the low field peak of the CT, (3381 mT) with two $\tau$*
*values (indicated in the figure). The assignment of the signals is given by the colored stripes*
*added to the spectra following the color code given on the complex structure given on the right.*
*The numbers next to the protons give the distances in Å extracted from the ENDOR doublets'*
*splitting. The asterisks mark the position of the blind spots.*
*3.1.2 Hahn echo decays*
Hahn echo decays were measured for all samples and could be well-fitted with a single
stretched exponential decay function (SE model) (eq. 1) :
$$y = A * exp\left(-\frac{2\tau}{T_m}\right)^{\beta} \tag{1}$$



For Gd-PyMTA, measurements were carried out in the range of 3-200 μM, and for Gd-TPMTA
the range was 25-200 μM; concentrations lower than 25 μM were not tested because of
sensitivity limits owing to the broader EPR spectrum of Gd-TPMTA. A few examples of echo-
decay data and their fits are shown in **Fig. S8.** The concentration dependence of $1/T_m$ measured
at 10 K on the CT for the PyMTA variants is given in **Fig. 4A**. We chose 10K because it is the
optimal temperature for DEER measurements considering the populations of the CT and the $T_m$
temperature dependence. (Goldfarb, 2014) We did not detect any apparent effect of the degree
of deuteration on $T_m$ and β, both of which show a clear concentration dependence. This
indicates that protons with hyperfine couplings in the range of 1-2 MHz (distance < 5 Å) do not
lead to decoherence as they may be within the nuclear spin diffusion barrier.(Wolfe, 1973)
$1/T_m$ of Gd-PyMTA is linearly dependent on concentration, [C], and the intercept of 0.05 μs$^{-1}$
gives $T_m(0)$=20 μs, this is $T_m$ free of SD contributions. The dependence of β on [C] is not linear,
reaching β=1 for [C]→0 (**Fig. 4B**). The dependence of $1/T_m$ on the magnetic field within the EPR
spectrum (10K and [C]=200 μM), shown in **Fig. 4C**, reveals the same dependence as reported
earlier(Raitsimring et al., 2014), where the central transition exhibits a longer $T_m$; a characteristic
of the tZFS mechanism (Raitsimring et al., 2014). The Hahn echo decay behavior of Gd-TPMTA,
presented in **Fig. 4D-F, was** generally like that of Gd-PyMTA, disclosing no dependence on the
deuteration levels. For Gd-TPMTA measurements, the concentration dependence
measurements were carried out at three field positions (1,2,3, see **Fig. 2A**) within the CT, and
the results of all three were practically the same; the data presented in **Fig. 4 D-F** corresponds to
position 1. For Gd-TPMTA $T_m(0)$=10 μs, the value of β is lower than for Gd-PyMTA and β=0.8 for
[C]→0. In general, β in the range of 1-2.5 suggests the presence of a fast dephasing process
attributed to SD or NSD(Salikhov et al., 1981; Eaton and Eaton, 2000), whereas β<1 is typical of
slow processes or is a signature of relaxation time distribution.(Salikhov et al., 1981)[30]
Accordingly, we attribute the reduction in β with concentration to a reduction in the SD
contribution, and the lower value of β at the low concentration limit of Gd-TPMTA is likely due
to a larger, more extensive distribution of relaxation times. The latter arises from the larger ZFS
and the significant contributions of transitions other than the CT at the CT field. We checked for
the effect of ID for Gd-PyMTA with [C]= 200 μM by measuring the echo decay as a function of
the length of the second pulse and found a negligibly small contribution to $1/T_m$ and β (see **Fig.**



**S9**). Therefore, we conclude that the contribution of ID to the concentration dependence of
$1/T_m$ is minimal.

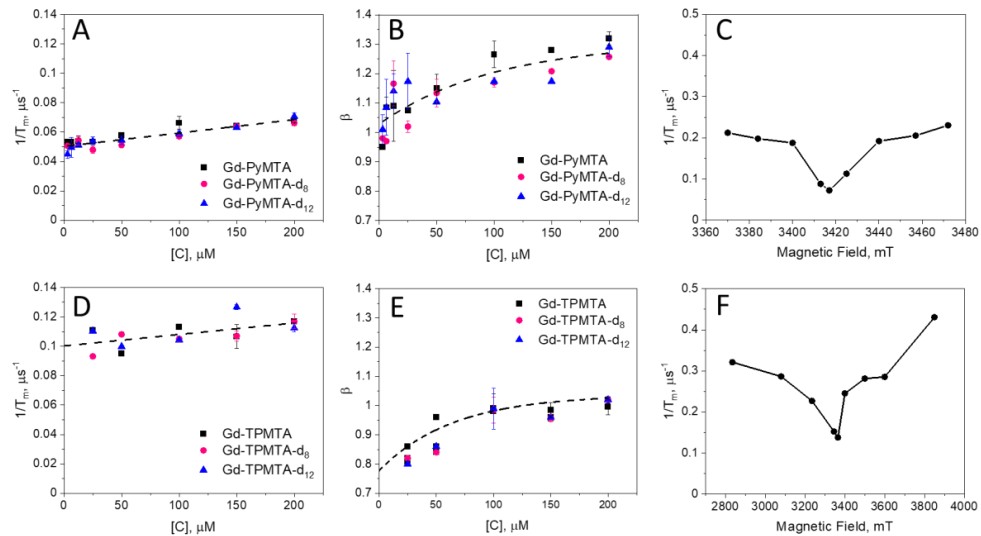

*Figure 4. The dependence of $1/T_m$ and $\beta$, measured at 10 K and the peak of the CT for the Gd-*
*PyMTA variants (A,B) and Gd-TPMTA variants (D,E) measured at position 1 given in Fig. 2A. The*
*dotted line in A and D is the linear fit with slopes of $9.07\times10^{-5}\pm8.2\times10^{-6}$ and $7.79\times10^{-5}\pm2.9\times10^{-5}$*
*$(\mu s,\mu M)^{-1}$ and intercepts of $0.05\pm0.01$ and $0.100\pm0.003$ $\mu s^{-1}$ for Gd-PyMTA and Gd-TPMTA*
*respectively. The dotted lines in B and E were obtained with an exponential function $y = y_0 +$*
*$A_1 e^{-\left(\frac{x}{T}\right)}$ to guide the eye. The parameters used were $y_0$=1.3, A=-0.28, T=100.8 $\mu s$ and $y_0$=1.04,*
*A=-0.26, T=64.5 $\mu s$ for Gd-PyMTA and Gd-TPMTA respectively. (C, F) The Field dependence of*
*$1/T_m$ for 200 $\mu M$ Gd-PyMTA (C) and 200 $\mu M$ Gd-TPMTA (D), measured at 10 K.*
Because the $T_1$ values of Gd(III) are relatively short and can influence its phase relaxation, we
carried out $T_1$ and $T_m$ measurements at different temperatures for Gd-PyMTA and Gd-TPMTA.
We measured only the non-deuterated variants as we did not see any effect of the complex
deuteration on the echo decays. The $T_1$ values were determined from inversion recovery
experiments and were analyzed using stretched exponents with values in the range of 0.7-0.8
(see **Fig. S10**). **Fig. 5A** shows the temperature dependence of $T_1$ and $T_m$ of 200 and 50 $\mu M$ Gd-
PyMTA; as expected $T_1$ is concentration independent. For Gd-TPMTA a broader range of
temperatures was accessed (1.6-15 K vs 10-20 K) and the results are given in **Fig. 5B**. For both
complexes, we found that, unlike $T_m$, $T_1$ was independent of the field position within the EPR
spectrum (**Fig. S11**); namely, it is the same for all Gd(III) EPR transitions. For Gd(III), we must
consider that it is not only the relaxation times that change with temperature but also the





relative populations of the various transitions. Accordingly, changes in the levels' populations
can influence the $T_m$ values measured at the CT. This effect is marginal for the temperature
range explored for Gd-PyMTA but can be significant for Gd-TPMTA below 7K.
To reveal the effect of $T_1$ on $T_m$, we plotted $1/T_m$ *vs* $1/T_1$, and the results are shown in **Figs. 5C-D**.
We observed a linear correlation for Gd-PyMTA (50 µM and 200 µM). For Gd-TPMTA, where a
wider range of temperatures was probed, a linear correlation was observed only for the 5-15K;
below 5K $1/T_m$ is fairly constant, indicating that the contribution of $T_1$ to the phase relaxation is
no longer significant.

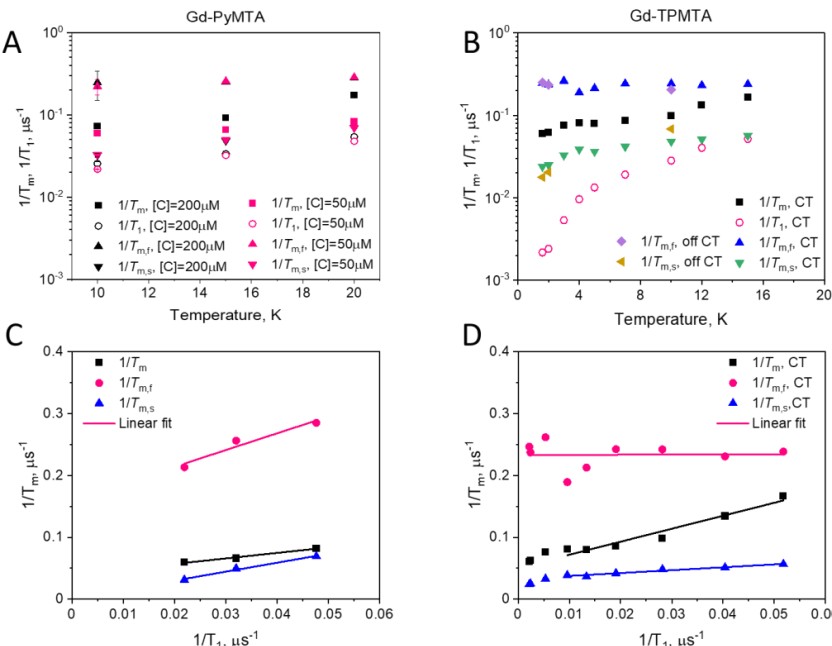

*Figure 5. The temperature dependence of $1/T_m$, determined with Hahn echo and CP trains, and*
*$1/T_1$ for Gd-PyMTA (A) and Gd-TPMTA (B). The dependence of $1/T_m$, determined with Hahn echo*
*and CP trains on $1/T_1$ for Gd-PyMTA (C) and Gd-TPMTA (D) along with the linear fit in the 6-20 K*
*range.*
The slopes of $1/T_m$ vs concentrations [C] for both complexes are the same within experimental
error **(Fig. 4)**. This indicates that the contribution of $1/T_{SD,ee}$ is negligible; otherwise a significant
difference would be expected because of the much broader EPR spectrum of Gd-TPMTA.
Therefore, we attribute the concentration dependence of $1/T_m$ to the indirect-T$_1$, $T_{SD,T1}$



mechanism, which is lineshape independent. As the two complexes have similar $T_1$ values,
similar slopes are expected. We used the known expressions for $T_{SD,T1,}$ and $T_{ID}$ to estimate their
theoretical contributions (See **Fig. S12** and the associated text) for Gd-PyMTA. We found the
predicted contribution of $T_{ID}$ is negligible, consistent with our experimental results, and that the
$T_{SD,T1}$ calculated without any fitting parameters reproduces the experimental data reasonably
(see **Fig. S12**), predicting a slope in the linear region of $1.2 \times 10^{-4}$ $\mu s^{-1}/\mu M$, compared to the
experimental slope $0.9 \times 10^{-4}$ $\mu s^{-1}/\mu M$. The over-estimated concentration dependence can result
from the non-exponential behavior of the echo decay and the inversion recovery, namely $\beta \neq 1$,
and the underestimation of $T_1$ determined by the inversion recovery sequence. The
contributions to $T_m(0)$ can be from the direct-$T_1$ relaxation, $T_{m,T1}$, residual NSD, and tZFS. As the
contributions of spin diffusion, being either NSD or SD, to phase relaxation can be suppressed by
DD, we proceeded with measurements of $T_m$ using CP trains to further resolve the various
contributions to phase relaxation.
*3.1.3 CP with n≤5*
To resolve the potential contribution of NSD induced by the very weakly coupled protons on the
tags to decoherence, we followed the approach used by Jeschke and coworkers(Soetbeer et al.,
2018), and measured the intensity of the last echo as a function of the interval between the
pulses for CP trains with *n*=2-5 refocusing pulses (see **Fig. 6**), while holding the time between the
first $\pi/2$ pulse and the last echo constant equal to $2\tau$. Interferences from overlapping stimulated
echoes can be eliminated by phase cycling up to *n*=5; beyond this, the phase cycle becomes too
demanding(Soetbeer et al., 2018) (the phase cycles used are listed in **Table S2**). The resulting
echo decays were analyzed using Eq.1 (examples of fits are shown in **Fig. S13**), and the data
from the protonated spin labels are given in **Fig. 7A**, where we plotted $1/T_m$ and $\beta$ as a function
of *n*, with *n*=1 corresponding to the Hahn echo. We observed the same general behavior for 200
and 25 $\mu M$ Gd-PyMTA; an initial significant decrease in $1/T_m$ from *n*=1 to *n*=2, followed by a mild
change between *n*=2 to *n*=4 and leveling off at n=5. $\beta$ exhibits a monotonic decrease from *n*=1
to *n*=3 and levels off at *n*≥3, where it reaches a value of 1. Nevertheless, the systematically
larger $1/T_m$ for 200 $\mu M$ than that for 25 $\mu M$ and the reduction of the differences between with
n, shows that DD can suppress SD for the 200 $\mu M$ sample, though for n=5 only partially. The
behavior of Gd-TPMTA is similar but less pronounced; $1/T_m$ decreases from *n*=1 to *n*=2 but then
levels off and $\beta$ levels off between *n*=3 and n=4. We find the suppression of NSD contributions





to be less likely because Gd-TPMTA has more weakly coupled protons on the label, and
therefore, the effect should have been larger for Gd-TPMTA, but the opposite was observed.
Such measurements, reported for Gd-DOTA-M in $D_2O$:glycerol-$d_8$ (25 μM) at 10 K at Q-band,
showed similar behavior, i. e. a mild decrease in $1/T_m$ and $\beta$ (Soetbeer et al., 2021b). $T_m$ reached
40 μs for n=5 for Gd-DOTA-M, compared to 29 μs for Gd-PyMTA at W-band.

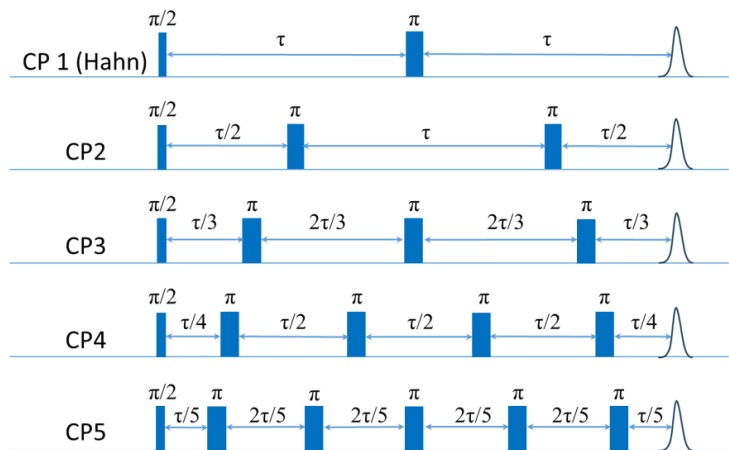

**Figure 6.** *The CP sequences for n=1-5. n=1 corresponds to the Hahn echo.*



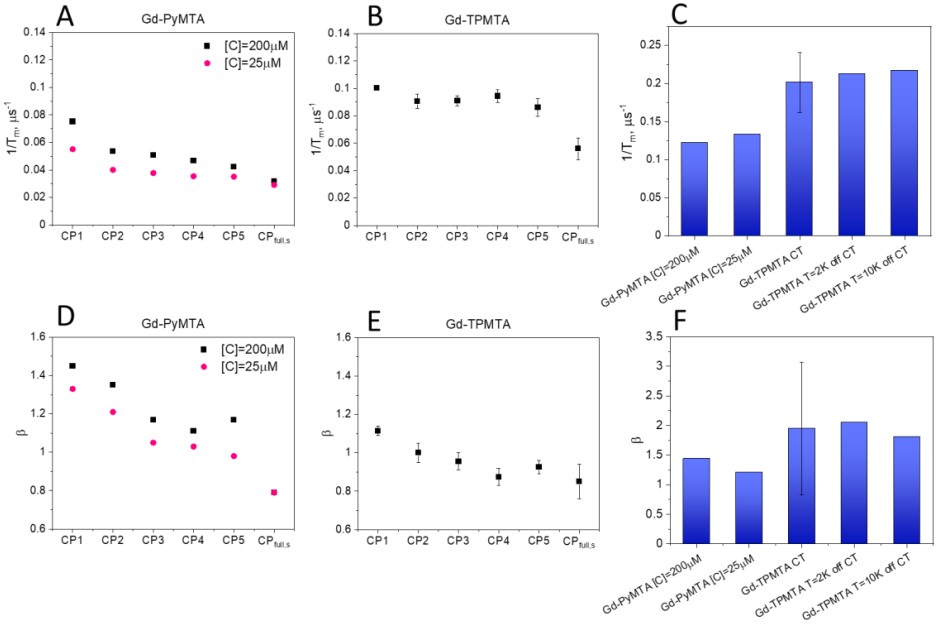

**Figure 7.** *Comparison of 1/T$_m$ measured at the CT and 10 K (A,B) and $\beta$ (D,E), determined by CP with n=1-5 and those of the slow component in the full CP train for Gd-PyMTA (A,D) and Gd-TPMTA (B,E). (C,F) Comparison of 1/T$_m$ of the fast component in the full CP train (C) and $\beta$ (F) for different complexes with different concentrations for Gd-PyMTA and Gd-TPMTA measured at different field positions within the EPR spectrum.*

*3.1.4 Full CP train*

The very mild effect of CP with *n*=5 on the phase relaxation of both complexes prompted us to improve the effectiveness of the DD by applying a CP train pulse sequence with a constant inter-pulse delay $\tau$ (see **Fig. 8A**) and the shortest available on our spectrometer (290 ns), to suppress potential contributions from fast processes to the phase relaxation. We refer to this as full CP train. An example of the echo trains produced by this sequence is given **Fig. 8B,** and the plot of the echo intensities as a function of time is presented in **Fig. 8C.** In this case, the data could not be satisfactorily fitted with a single stretched exponent and a sum of two stretched exponents (termed earlier as SSE (Soetbeer et al., 2018)) was used as follows:

$$y = A * exp\left(-\frac{t}{T_{m,f}}\right)^{\beta_f} + (1 - A) * exp\left(-\frac{t}{T_{m,s}}\right)^{\beta_s} \tag{2}$$



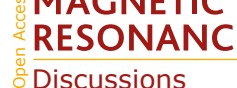

where $t$ is the time between the first $\pi/2$ pulse and the observed echo, and the subscripts $f$ and
$s$ correspond to fast and slow processes.
As mentioned earlier, CP measurements with pulses that are not ideal because of their small
bandwidth compared to the EPR spectral width produce echoes that are not pure refocused
echoes but have contributions from stimulated echoes that decay with some combination of $T_1$
and spectral diffusion (Kurshev and Raitsimring, 1990; Mitrikas, 2023). To ensure that the
observed SSE analysis is not a consequence of the contributions of such unwanted echoes, we
performed a series of calculations presented in the SI (**Figs. S14-S16**). These show that the
stimulated echo contribution leads to overestimation of $T_m$ by no more than 20% and that a
single stretched exponential function can fit the calculated echo intensities. To further ensure
that the two observed components derived from the experimental results are not a
consequence of artifacts in the applied pulse sequence, we carried out similar measurements on
a nitroxide (MTSL) spin label in $D_2O$:glycerol-$d_8$ (25 μM), and the results are shown in **Fig. S17**. In
this case, the echo train could be fitted well with only one stretched exponent. Therefore, we
concluded that the two resolved populations are intrinsic to the Gd(III) complexes studied and
are not a consequence of the stimulated echo contributions or experimental artifacts.

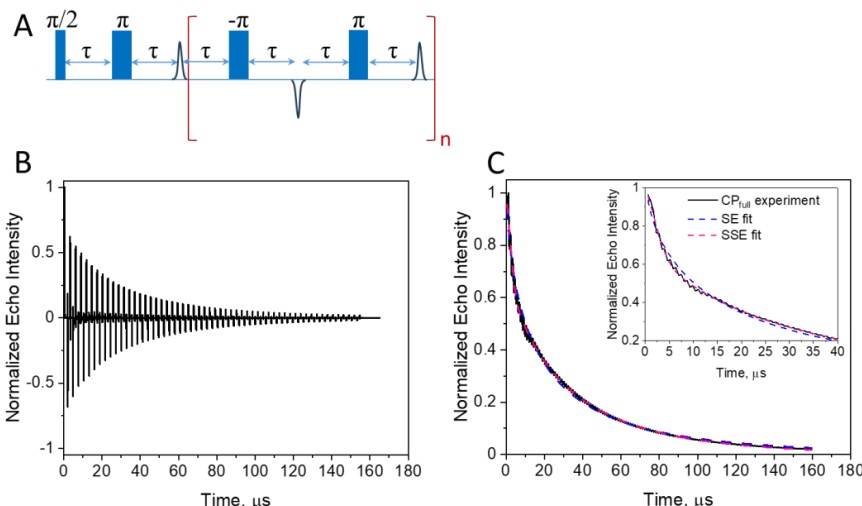

**Figure 8**. (A) The CP (n=137, $\tau$=290 ns) sequence applied and (B) the resulting echo train for Gd-
PyMTA, 50 μM, measured at 10 K and the maximum of the CT (C) The plot of the integrated echo
intensity of the individual echoes as a function of time with the data fit using a single stretched
exponential and a sum of two stretched exponentials. The inset shows an expanded part of the



*trace, highlighting the fit differences. The fitting parameters were: $T_m$=32 μs, β=0.58, A=1.1 for SE*
*fit and $T_{m,f}$=3.3 μs, $β_f$=1.85, $A_f$=0.25, $T_{m,s}$=29.6 μs, $β_s$=0.78, A=1.1 for SSE fit, respectively.*
**Fig. 9** summarizes the dependence of the slow and fast CP decay rates, $1/T_{m,s}$ and $1/T_{m,f}$ , and
the associated $β_s$ and $β_f$ on concentration and temperature for all Gd-PyMTA and Gd-TPMTA
variants. The slow component of Gd-PyMTA, which has a contribution of 75-80%, is
concentration and deuteration independent, and at 10 K $T_m$ is 35 μs, about twice as long as that
measured by the Hahn echo decay (18 μs), well beyond the 20% expected overestimation. The
same holds for β, which is reduced to about 0.75. For comparison with the values obtained with
$n$=1-5 we added the data to **Fig. 7A,D.** Interestingly, while CP with $n$=5 could not eliminate the
concentration dependence, the entire train did ($n$~140). Here, the $1/T_{m,s}$ value for 25 and 200
μM Gd-PyMTA coincided. We should bear in mind that the full CP train and CP $n$=2-5 are
different types of experiments; for the former, the number of π pulses and τ are held constant
and the recorded signal is the intensity of the occurring refocused echo after each π pulse, and
for the latter the number of pulses is constant, τ is varied and the intensity of the last echo is
measured. This might be why full CP train refocuses the SD contributions better. There is an
increase in $1/T_{m,s}$ with temperature, whereas $β_s$ and the relative population remain constant. For
Gd-TPMTA, as for Gd-PyMTA, the slow component is not dependent on concentration nor
deuteration level. Still, there is an increase of $β_s$ and a decrease in its relative population with
increasing temperatures. The values of $1/T_{m,s}$ and $β_s$ at 10 K are added to **Fig. 7B,E** for
comparison with those obtained for $n$=1-5. The dependence of $1/T_{m,s}$ on the temperature and
$1/T_1$ for the two complexes are shown in **Fig. 5.**
For the fast component, the spread of the data points is quite large for all variants, and no
systematic variation in concentration nor deuteration is observed. Here, $T_{m,f}$ is 3-6 μs and $β_f$=1.3-
3; no temperature or concentration dependence was detected within the experimental error. A
comparison of the various values of $1/T_{m,f}$, and $β_f$ for different samples and temperatures is
given in **Fig. 5C,F.** The relative contribution of the two components is fairly constant in the
temperature range tested for Gd-PyMTA, whereas for Gd-TPMTA a significant increase in the
contribution of the fast component with increasing temperature is observed in the range of 6-15
K **(Fig. 10)**.

**MAGNETIC RESONANCE**
Open Access Discussions

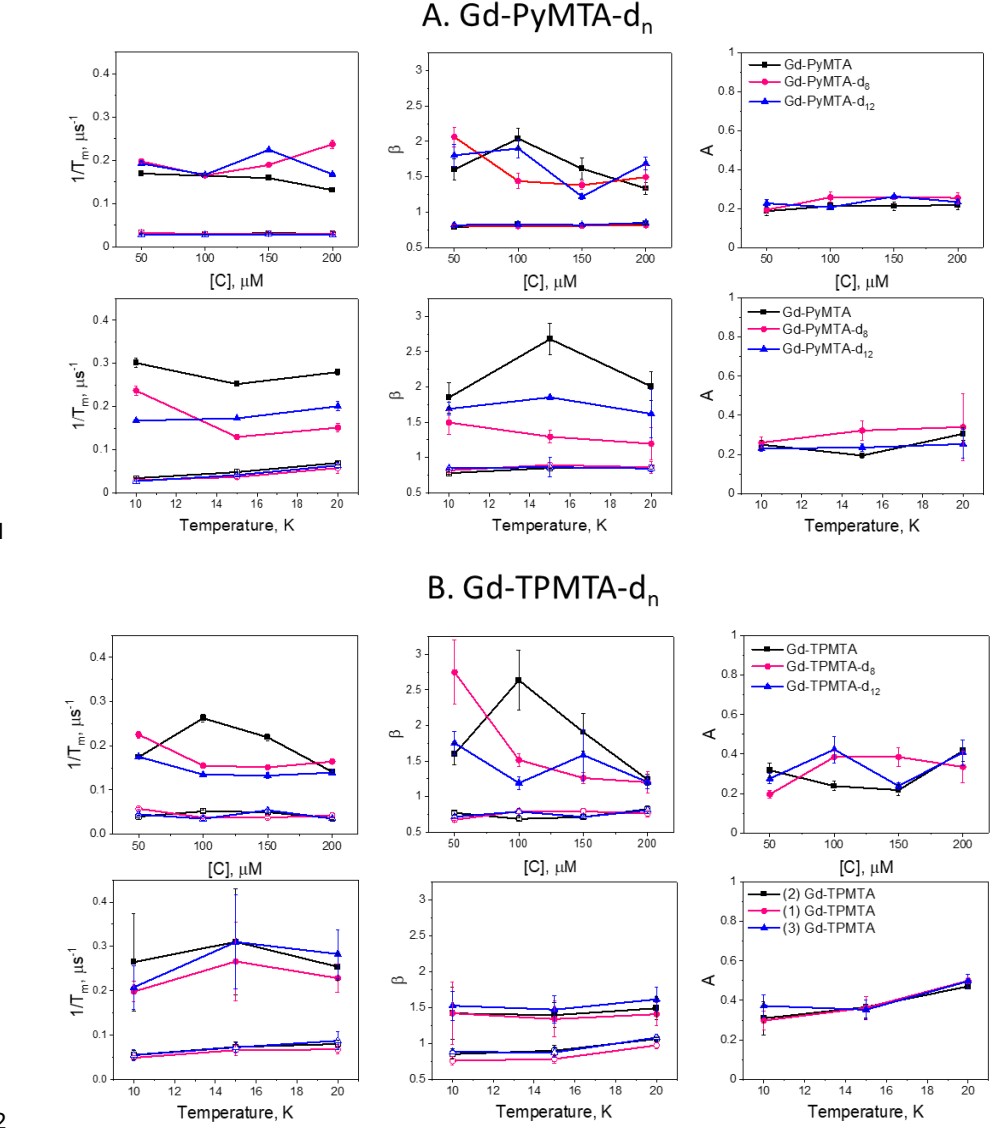

*Figure 9. The dependence of the slow (open symbols) and fast (filled symbols) CP full train decay rates, the associated stretched exponent and their relative population measured for: (A) The Gd-PyMTA samples measured at the peak of the central transition as a function of concentration (10 K) and as a function of temperature for 200 µM. (B) The Gd-TPMTA samples measured at position 1 in the central transition as a function of concentration (10 K) and as a function of temperature for 200 µM Gd-TPMTA at field positions 1,2,3. The field positions are defined in **Fig. 2.***





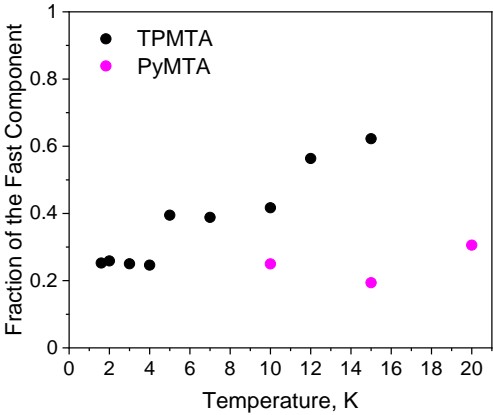

*Figure 10. The dependence of the relative contribution of the fast component, measured at the*
*CT, as a function of temperature for Gd-PyMTA (magenta) and Gd-TPMTA (black).*
We also investigated the field dependence of the decay rates of two components at two
temperatures, 10 and 20 K (**Fig. 11**). For the Hahn echo, we observed a clear enhancement of
the decay rate outside the CT; in contrast, the slow component showed a minimal change across
the CT at both temperatures. Also, the difference between the CT and the other transitions was
significantly weaker for the fast component than for the Hahn echo. Interestingly, the contrast
between the CT and the other transitions is manifested in the two components' relative
contributions. For both complexes, the contribution of the fast component is lower at the CT
than outside the CT and the contribution of the fast component increases with temperature in
all fields.



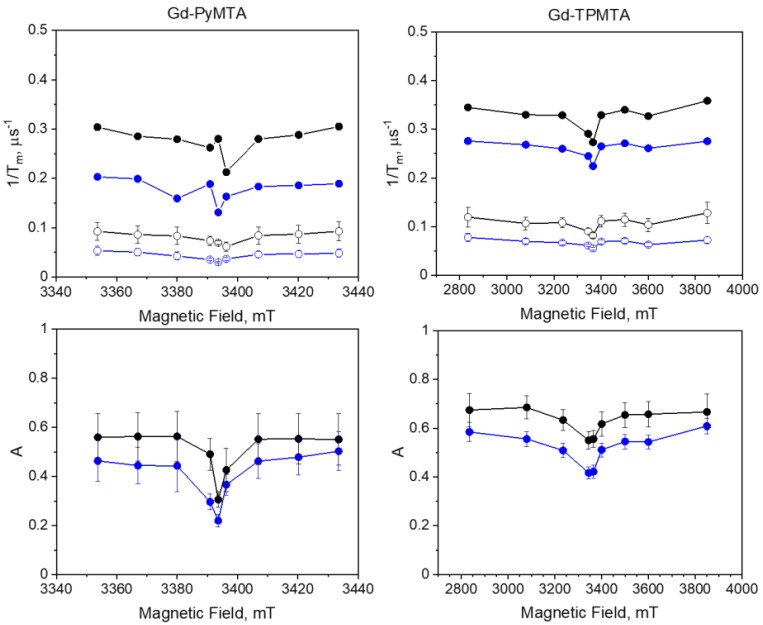

**Figure 11.** *The field dependence of $1/T_m$ for the slow and fast components at 10 and 20 K (top) and of the relative contribution of the fast component, A (see Eq. (2)), (bottom) for both complexes.*

From the full CP train measurements, we conclude that (i) two populations of spins with different dominating phase relaxation mechanisms are observed for the two complexes. (ii) Any residual $SD_{ee}$ and $SD_{T1}$ contributions are suppressed under the CP train conditions. (iii) We tentatively assign the dominating mechanism that governs the slow-relaxing population to the direct $T_1$ mechanism and tZFS for the fast-relaxing population. (iv) The relative contribution of the tZFS mechanism is lower at the central transition than at the other transitions. (v) Gd-TPMTA, which has a significantly larger ZFS than Gd-PyMTA, has a larger population dominated by the tZFS, which is also temperature-dependent. The temperature dependence does not follow the relative intensity of the CT with temperature and, therefore, suggests that the tZFS fluctuations increase with temperature.



### 3.2. Influence of protein deuteration.
For nitroxide spin labels, protein deuteration increases $T_m$, by a factor of ~4 (Ward et al., 2010;
Schmidt et al., 2016). To see if Gd(III) spin labels experience the same effect, after exploring the
phase relaxation behavior of the free Gd-PyMTA and Gd-TPMTA spin labels in deuterated
solvents, we proceeded to examine their phase relaxation after their attachment to protonated
and deuterated proteins in deuterated solvents. Ubiquitin D39C/E64C was labeled with Gd-
PyMTA and Gd-TPMTA, producing doubly labeled proteins typically used for DEER applications.
The concentrations were ~25 μM and ~50 μM for Gd-PyMTA and Gd-TPMTA labeled ubiquitin,
respectively. The Hahn echo decays were fitted using Eq.1, as was done for the free labels
(examples are shown in **Fig. S18**), and the results are summarized in **Fig. 12** for measurements
on the CT at 10 K. The attachment to $^1$H-ubiquitin increased $1/T_m$ by a factor of 2.4 for Gd-
PyMTA and 1.8-1.9 for Gd-TPMTA, with no significant effect on the degree of label deuteration.
While protein deuteration led to a slight decrease of $1/T_m$ (~10%) for Gd-PyMTA, for Gd-TPMTA
labeled ubiquitin, a significant effect was noticed only for Gd-TPMTA-$d_{12}$, almost reaching the
value of the free label. The dependence of β on protein deuteration is insignificant for Gd-
TPMTA. In contrast, for Gd-PyMTA, a gradual decrease with the increased degree of the label
deuteration is observed for the deuterated protein, where for Gd-PyMTA-$d_{12}$ it reaches β=1, as
for the free label. A small effect of the protein deuteration was also observed for the Gd-DO3A
labeled ubiquitin (50 μM) (**Fig. 12**). The low impact of protein deuteration on the Gd(III) $T_m$
values compared to nitroxide can be attributed to the Gd(III)'s much shorter $T_1$, which provides
the upper limit to $T_m$ ($T_m \leq 2T_1$).
To further explore the origin of the significant reduction in the phase relaxation rate while
bound to protein and the small effect of protein deuteration, we carried out full CP train
measurements on the protonated and deuterated protein samples for Gd-PyMTA and Gd-
TPMTA. Like the free labels, the data could not be fitted well with one stretched exponent, and
a sum of two such exponents was needed, one with a fast decay and the other with a slow
decay **(see Fig. S19).** The results for the population with the slow decay for protonated and
deuterated ubiquitin with protonated Gd-PyMTA are shown in **Fig. 12**. The difference between
the protonated and deuterated proteins was small. Same as in the case of the free label, we
observed a reduction of about a factor of 2 in $1/T_m$ for the slow population under CP train
conditions, as compared to the value for Hahn echo. Interestingly, $1/T_m$ in the free Gd-PyMTA in



a deuterated solvent is smaller by a factor of about two compared to the protein value. The
same behavior was observed for Gd-TPMTA.
What is the source of the faster phase relaxation in the protein compared to the free complex?
It cannot be attributed to the direct $T_1$ mechanism because of the similar $T_1$ values, 50.5 µs for
protein and 45.5 µs for free complex. One possibility could be NSD from the non-deuterated
HEPES molecules used as buffer, which results in 2.5% protons in the solvent. Another possibility
could be the lower amount of glycerol in the protein samples (8:2 v/v vs 1:1 v/v for free Gd-
PyMTA). To check this possibility, we prepared solutions of Gd-PyMTA in 15 mM HEPES in 8:2
v/v vs 1:1 v/v $D_2O$/glycerol-$d_8$ and measured their Hahn-echo decays at 10K. The results, given in
**Fig. S20**, show that the contribution of the protonated HEPES is small, but that of the lower
amount of glycerol is significant. These two effects account only for about 80% of shorter $T_m$ in
the protein. An additional contribution can come from the fact that the proteins are doubly
labeled, i.e., every Gd(III) center has a neighbor ~4.2 nm away from it (see **Fig. S21**). Accordingly,
its phase relaxation can be affected by indirect $T_1$ due to the neighbor, which is concentration-
independent. The relaxation of this neighbor is responsible for the Gd(III)-Gd(III) RIDME (The
Relaxation-Induced Dipolar Modulation Enhancement) PD-EPR experiment, where it undergoes
both single, double and triple quantum flips during the mixing time.(Razzaghi et al., 2014) In this
case, mutual Gd(III) pair flip-flops can also induce relaxation.(Tyryshkin et al., 2012)



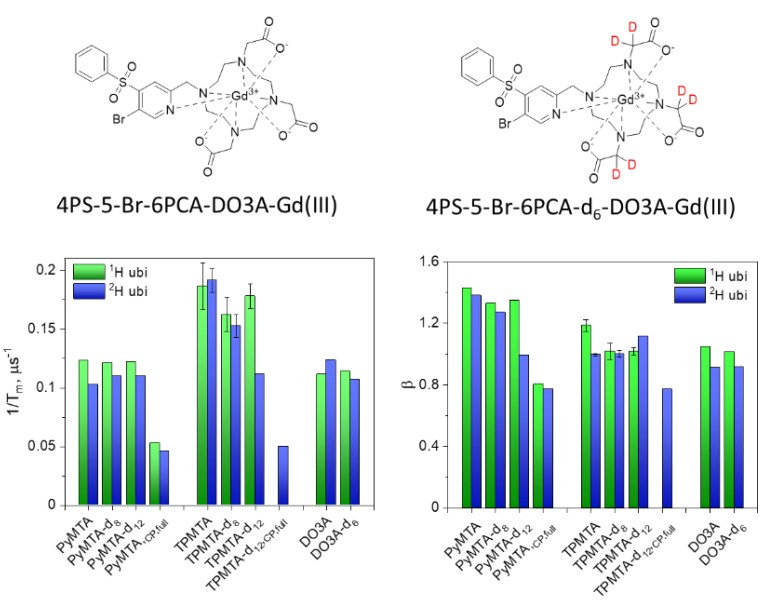

**Figure 12.** *The structure of the Gd-DO3A label and the summary of the Hahn echo $1/T_m$ and $\beta$ of*
*ubiquitin labeled with Gd-PyMTA, Gd-TPMTA, and Gd-DO3A with various degrees of deuteration.*
*Gd-PyMTA, the results for the CPMG slow component have also been added.*
**Conclusions.**
In this work, we explored the mechanisms responsible for the phase relaxation of Gd(III) spin
labels at 95 GHz, exploring whether deuteration of the label or the protein can extend the phase
relaxation. To resolve the relaxation mechanisms, we first studied the free label with different
degrees of deuterations in a deuterated solvent and examined both concentration and
temperature dependencies. We compared two labels having very different ZFSs, which helped
resolve various relaxation mechanisms. $T_m$ was determined from both Hahn echo decay and CP
echo trains. Our conclusions are as follows:
1.  Protons with hyperfine couplings in the 1-2 MHz range, situated at a distance <5 Å, do
not affect $T_m$ and are located within the nuclear spin diffusion barrier.
2.  In the range of 5-200 µM the concentration dependence of the free tag is primarily
determined by the indirect $T_1$- induced mechanism.
3.  At the limit of [C]$\rightarrow$0, the contributions to $T_m$ (0) can be residual NSD of the protons on
the pyridine rings with hyperfine couplings below 0.4 MHz, tZFS, and direct $T_1$. Since

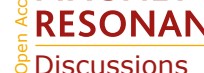

$T_m(0)$ for Gd-TPMTA is shorter by a factor of about 2 and $T_1$ values of both complexes are similar, we attribute the difference to the increased contribution of residual NSD or tZFS. Gd-TPMTA has 12 weakly coupled protons vs 7 for Gd-PyMTA, and its ZFS is significantly larger. The CP measurements with $n$=2-5 had a more substantial suppression effect on $T_m$ for Gd-PyMTA, suggesting that it originated from SD due to electron-electron interactions and that NSD was not suppressed under these conditions (n=5), thus leaving tZFS and direct-$T_1$ as the mechanisms governing $T_m(0)$.

4.  Full train CP measurements ($n$~140) resolved the presence of two populations: One with a slow phase relaxation and the other with a fast one. The dominating mechanism for the slow population is direct-$T_1$. Its $T_m$ showed no concentration dependence and was longer by a factor of about 2 relative to the Hahn echo decay for both complexes, yet keeping their relative values. We tentatively assign the decrease in $1/T_{m,s}$ to full suppression of the residual indirect $T_1$-induced and NSD mechanism, made possible by the relatively short $\tau$=290 ns used in the full train. This is supported by the more significant difference between $n$=5 and the full train for Gd-TPMTA, which has more distant protons.

5.  For the fast relaxing population, $1/T_{m,f}$ is larger for Gd-TPMTA; therefore, we assign it to populations where the tZFS dominates, supported by its more extensive field dependence than $1/T_{m,s}$.

6.  Because of the relatively short $T_1$ and the contribution of the tZFS mechanism, protein deuteration does not significantly affect $T_m$. The shorter $T_m$ for the doubly labeled proteins is attributed primarily to the lower glycerol amount in the sample and indirect-$T_1$ owing to the presence of a close-by Gd(III) neighbor.

The above shows that prolonging $T_m$ would require increasing $T_1$, which can be achieved by lowering the temperature. However, this will be at the expense of the CT population, thus reducing sensitivity in DEER measurements. Another option is to reduce the spectrometer frequency, which again will cause broadening of the central transitions and impede sensitivity. Yet another way to increase $T_m$ is to choose a label with a smaller ZFS (Ossadnik et al., 2025). A very small ZFS, however, introduces significant difficulties in analyzing DEER data for distances below 3 nm (Dalaloyan et al., 2015).



**Code and data availability:** All data reported herein and the EasySpin scripts for simulation of
TPMTA tag ED-EPR spectrum can be accessed at https://doi.org/10.5281/zenodo.15112854
**Author contribution**:    EE-investigation, visualization, writing – review and editing. XZ-
investigation, visualization CC-investigation. YY-investigation. AB-formal analysis, methodology,
writing – review and editing. GM -formal analysis, visualization, writing – review and editing XS-
Conceptualization, funding acquisition, supervision, writing – review and editing.    DG -
Conceptualization, funding acquisition, supervision, writing – original draft preparation.
**Competing interests** : DG is a member of the executive editorial board of Magnetic Resonance.
**Acknowledgments.**
The work was funded by the Joint NSFC (China) -ISF grant (No. 3559/21) and was made possible
in part by support from the Helen and Martin Kimmel Institute for Magnetic Resonance
Research and the historic generosity of the Harold Perlman Family (D. G.).

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
