# Peer review of "Can label or protein deuteration extend the phase relaxation time of Gd(III) spin labels?"

_Magnetic Resonance, 2025_

## Referee Comment (RC2)

Referee Comment Manuscript 10.5194.mr/mr-2025-6

The manuscript represents an interesting example of understanding to design and design for understanding related to the impact of the deuteration within the context of the relaxation processes. The message of the manuscript is clear and exhaustively delivered; however, for fulfilling the main 'take-home' message of the manuscript, some minor points can be revised:

- For the comparison of the values of $1/T_{m,f}$ and $\beta_f$ for different samples and temperatures (figure 7 C, F; please notice into the text at page 20 such figure has been referred as '5'), the authors refer to Figure 10 (page 22) for describing the contribution of the fast component. Besides the *fairly constant behaviour* for the PyMTA, it would be interesting to provide further elements to the discussion on the behaviour of TPMTA, exhibiting a completely different behaviour.

- The general approach proposed does not mention the effect of the pH, which may have an impact into the affinity of the two main ligands described; such an effect on the relaxation is probably beyond the scope of the manuscript, but it can be worth to mention also that tuneable parameter (i.e., pH).

- The assignment of dominating mechanism assigned for the two populations (*slow* and *fast*), as summarized on page 23 (lines 8-11) can eventually be reinforced by citing known structures where the $T_1$ and tZFS are distinctively contributing to the relaxation paths. It may support the effect of the deuteration for 'small' molecules and validate the less pronounced effect on labelled proteins.

- Please notice that the authors refer to Figure 2D (page 10) but the capital letter on the figure 2 (page 11) is missing. A-B-C-D on the four panel must be revised.

---

## Referee Comment (RC3)

**General comments**

Thanks to a detailed comparative analysis of the relaxation data of two Gd(III) complexes with different zero-field splitting parameters, the authors rationalise their finding that deuterating the protein only results in a very limited enhancement of the phase memory time, a crucial parameter in pulse dipolar spectroscopy measurements.

The manuscript is clearly written. The adopted systematic multi-technique approach to isolate the various contribution to electron spin decoherence, the provided data and the data analysis fully support the conclusions.

Below are some comments for the authors.

**Specific comments**

**Introduction**

- Page 3, line 14: although the authors clearly specify what the terms "*direct spin-lattice relaxation*" and "*indirect T*1" refer to, in the context of spin-lattice relaxation mechanisms "direct" may be misinterpreted as the contribution to  $1/T_1$  which is proportional to the temperature ( $(1/T_1)_{dir} = A_{dir}$ -T).
- Page 5, line 21: "*This showed that the protein's protons do affect phase relaxation*". I find it difficult to follow as the paragraph mostly refers to the deuteration of the solvent.
- Page 5, line 31: "the complexes were dissolved in  $D_2O/glycerol-d_8$ , thus serving as a reference for the longest possible  $T_m$ ". Was the deuteration of the solvent assessed?

**Experimental**

- Page 6, line 22: "*The synthesis of 4PS-5-Br-6PCA-(dn)-DO3A-Gd(III) is described in detail in the Supporting Information*". As the corresponding structure is not reported in Figure 1, it may be useful to refer the readers to Figure 12.
- Page 8, lines 23-25: according to Raitsimring et al., 2014 *it was observed that in measurements which require a time base exceeding 12–13 µs the phase of the output echo signal sometimes varied substantially due to some features of the 'Quinstar' power amplifier. When a larger time base was required, the measurements were performed point by point using manual phase correction to achieve a maximum echo amplitude. Was a similar issue found in the measurements for this work?*
- Page 9, line 2: why was the inversion recovery sequence chosen over other methods less prone to spectral diffusion?
- Page 9, line 12: which version of the EasySpin program package was used?
- Page 9, lines 13-14: "The distributions of ZFS parameters were considered using a built-in EasySpin functionality (DStrain parameter)". Were the distributions of the ZFS parameters D and E considered to be uncorrelated or correlated?
- Page 9, lines 19-20: does the reported equation assume that the echo is generated by two pulses with same amplitude or same length? I went through the cited reference (Raitsimring et al., 2013) but I am afraid I didn't manage to locate the renormalisation equation in the form stated in the manuscript.

**Results and discussions**

- Page 11, figure 2: please, check the labels (D is missing and the figure next to A has none).
- Page 11, figure 2: "*small % of free*". What does this refer to? Does the component also appear in the inset of panel B?
- Page 12, lines 1 5: it may be worth specifying in the main text that the coupling constants have been converted into distances under the assumption of a purely dipolar coupling. This is already mentioned in the caption of Table S1.
- Page 12, line 17 (equation 1): I think that the correct form of the equation should read  $y = 4 * ern \left[ -\left(\frac{2\tau}{2}\right)^{\beta} \right]$

$$y = A * exp\left[-\left(\frac{2\tau}{T_m}\right)^r\right]$$

- Page 15, figure 5: especially considering the number of points, it may be good to have the corresponding data in the SI.
- Page 15, figure 5: what are the corresponding β values?
- Page 16, lines 18-19: "while holding the time between the first  $\pi/2$  pulse and the last echo constant equal to  $2\tau$ ". It may be better to reformulate this as  $\tau$  is not constant throughout the experiment.
- Page 16, lines 26-27: "β exhibits a monotonic decrease from n=1 to n=3 and levels off at n≥3, where it reaches a value of 1". β for 200 μM Gd-PyMTA seems to be consistently above 1 (Fig. 7D) whereas β for Gd-TPMTA seems to reach an asymptotic value < 1 (Fig. 7E).</li>
- Page 18, Figure 7: it may be better to report the concentration of the Gd-TPMTA sample.
- Page 20, line 2 (caption of Figure 8): the reported parameters do not seem to be related to Equation 2 (Af + A ≠ 1, no reference to Af in Equation 2).
- Page 20, line 16: "This might be why full CP train refocuses the SD contributions better". What does "This" refer to? E.g.,  $\tau = \tau_{min}$  in the full CP train experiment? Were full CP train traces recorded for  $\tau$  values larger than  $\tau_{min}$  to check that indeed  $\tau = \tau_{min}$  yields the best refocussing conditions?
- Page 22, Figure 10: what were the used concentrations?
- Page 22, Figure 10: why is there no datapoint for Gd-TPMTA at 20 K if this experiment has been reported in Fig. 9B (bottom right)?
- Page 22, lines 5-6: "For the Hahn echo, we observed a clear enhancement of the decay rate outside the CT". Is it worth referring the reader to Fig. 4 C/F?
- Page 23, Figure 11: what do the colours and the filled/empty symbols refer to?
- Page 23, Figure 11: were the pulse amplitudes adjusted for the individual field positions? (see *e.g.* the reported equation in Section 2.4).
- Page 25, lines 10-11: "the contribution of the protonated HEPES is small, but that of the lower amount of glycerol is significant". How was the effect of the lower amount of glycerol interpreted? Enhanced instantaneous diffusion due to a poorer quality of the glass?
- Page 26, Figure 12: to enable direct comparison, it may be useful to include the reference values (unconjugated tag) to this plot.
- Page 26, line 4 (caption of Figure 12): "*Gd-PyMTA, the results for the CPMG slow component have also been added*". I am afraid I can't follow what this statement refers to: CP data for Gd-TPMTA have been reported as well. Moreover, from what I understood the Meiboom-Gill variant of the CP sequence was not performed.

**Supplementary information**

- Page S15, Figure S9B: why were the  $1/T_m$  data plot against  $t_{\pi}$ ?  $(1/T_m)_{ID}$  is expected to be proportional to  $\sin^2(\theta_2/2)$ , see Equation S1. As a related point, are the experimental results consistent with the theoretical prediction?
- Page S21: "As shown by Mitrikas and co-workers". Does this reference to Mitrikas 2023?
- Page S24, caption of Figure 20: it would be beneficial to specify in the caption what the sample labels (*e.g.*, Gd-PyMTA HEPES 8:2) refer to.

**Technical corrections**

- Page 9, line 1: " $T_1$  measurements **for** were performed using the inversion recovery sequence,  $\pi t_{wait} \pi/2 \tau \pi \tau echo$ , with varying  $t_{wait}$ ". Please, remove the "for".
- Page 9, line 29: \*10K → 10 K (add a space between the figure and the units). This occurs
  in other places throughout the text.
- Page 11, line 14: \**phenyl rings*  $\rightarrow$  phenyl ring (GdPyMTA has only one phenyl ring)
- Page 13, line 15: where the central transition exhibits a longer  $T_m$ ; a characteristic of the *tZFS mechanism*. It may be better to replace the semicolon with a comma.
- Page 14, line 7 (caption of Figure 4):  $(\mu s, \mu M)^{-1} \rightarrow (\mu s \cdot \mu M)^{-1}$  or  $\mu s^{-1}/\mu M$ .
- Page 14, line 9 (caption of Figure 4): in the equation for the dotted lines in panels B and E the *x* axis is a concentration; the parameter *T* should have the same units.
- Page 15, line 6: the sentence "*a linear correlation was observed only for the 5-15K*;" is incomplete.
- Page 21, Figure 9B, bottom right: do the numbers in brackets refer to the different deuteration conditions?
- Page 24, line 14: \*for Gd-TPMTA labelled ubiquitin, a significant effect was noticed. I would consider removing the comma.
- SI, page S14, notes to Table S1: \*Gd-F distance  $\rightarrow$  Gd-H distance.
- SI, page S15, Figure S10: for the sake of a better presentation it may be more convenient to estimate the magnetisation at the thermal equilibrium by averaging a few points at the end of each trace; this would ensure that all the traces have 1 as their asymptotic value.
- SI, page S16, Figure S11: as both  $T_1^{-1}$  and  $T_m^{-1}$  are reported on the same plot, it may be better to use a logarithmic *y* scale.
- SI, page S16, beginning of section 7: \**the contributions of*  $T_{ID} \rightarrow$  the contribution of  $T_{ID}$ .
- SI, page S20: multiple spacing in \**pulses are* so close.
- SI, page S23, caption of Figure S19:  $*A_f \rightarrow A$  to keep consistency with Equation 2 in the main text.

---

## Author Comment (AC1)

**Response to Collauto comments**

General comments Thanks to a detailed comparative analysis of the relaxation data of two Gd(III) complexes with different zero-field splitting parameters, the authors rationalise their finding that deuterating the protein only results in a very limited enhancement of the phase memory time, a crucial parameter in pulse dipolar spectroscopy measurements. The manuscript is clearly written. The adopted systematic multi-technique approach to isolate the various contribution to electron spin decoherence, the provided data and the data analysis fully support the conclusions. Below are some comments for the authors.

Thank you for your positive evaluation and comment, and for reading our manuscript extensively.

Specific comments

Introduction

• Page 3, line 14: although the authors clearly specify what the terms "direct spin-lattice relaxation" and "indirect T1" refer to, in the context of spin-lattice relaxation mechanisms "direct" may be misinterpreted as the contribution to 1/T1 which is proportional to the temperature $((1/T1)_{dir} = A_{dir} \cdot T)$.

We think that our definitions should help avoiding such misunderstandings.

• Page 5, line 21: "This showed that the protein's protons do affect phase relaxation". I find it difficult to follow as the paragraph mostly refers to the deuteration of the solvent.

We rephrased the sentence to "These results showed that the protein's protons affect the phase relaxation of the Gd(III) spin label. Thus, one would expect that deuteration of the protein should help reduce the decoherence." Hope it helps.

• Page 5, line 31: "the complexes were dissolved in D2O/glycerol-d8, thus serving as a reference for the longest possible Tm". Was the deuteration of the solvent assessed?

No.

Experimental

• Page 6, line 22: "The synthesis of 4PS-5-Br-6PCA-(dn)-DO3A-Gd(III) is described in detail in the Supporting Information". As the corresponding structure is not reported in Figure 1, it may be useful to refer the readers to Figure 12.

This would be a problem as the call to the figs should be in consecutive numbers and this would require moving fig 12 up, which will disturb the flow of the manuscript.

• Page 8, lines 23-25: according to Raitsimring et al., 2014 it was observed that in measurements which require a time base exceeding 12–13 µs the phase of the output echo signal sometimes varied substantially due to some features of the 'Quinstar' power amplifier. When a larger time base was required, the measurements were performed point by point using manual phase correction to achieve a maximum echo amplitude. Was a similar issue found in the measurements for this work?

No. We checked manually for the phase of the echo at short and long times, and no variations were detected. The spectrometer has improved since 2014.

• Page 9, line 2: why was the inversion recovery sequence chosen over other methods less prone to spectral diffusion?

Indeed, the inversion recovery sequence is more prone to spectral diffusion than saturation recovery. As the focus of the work is the phase memory times, and all $T_1$ measurements were done in the same way for comparison purposes we think that this does not affect any of the conclusions of the work.

• Page 9, line 12: which version of the EasySpin program package was used?

Easyspin version 6.0.0-dev.50, this was added to p.9.

• Page 9, lines 13-14: "The distributions of ZFS parameters were considered using a built-in EasySpin functionality (DStrain parameter)". Were the distributions of the ZFS parameters D and E considered to be uncorrelated or correlated?

The distributions in D and E were treated as uncorrelated. This was added to page 9.

• Page 9, lines 19-20: does the reported equation assume that the echo is generated by two pulses with same amplitude or same length? I went through the cited reference (Raitsimring et al., 2013) but I am afraid I didn't manage to locate the renormalisation equation in the form stated in the manuscript. Results and discussions

Eq. (6) of the referenced paper, taken with the angles $\theta_{1,0} = \pi\alpha/2$; $\theta_{2,0} = \pi\alpha$; $\theta_{3,0} = 0$, corresponding to our experiment (two-pulse Hahn echo); in combination with the well-known transition matrix element between the $m_S$ and $m_S+1$ levels, gives the equations given in the text.

• Page 11, figure 2: please, check the labels (D is missing and the figure next to A has none).

Was corrected.

• Page 11, figure 2: "small % of free". What does this refer to? Does the component also appear in the inset of panel B?

The zoom insert was changed to the correct one. There is no contribution of free label.

• Page 12, lines 1 – 5: it may be worth specifying in the main text that the coupling constants have been converted into distances under the assumption of a purely dipolar coupling. This is already mentioned in the caption of Table S1.

This was added to the caption of Fig. 3.

• Page 12, line 17 (equation 1): I think that the correct form of the equation should read $y = A * exp\left[-\left(2\tau/Tm\right)^\wedge \beta\right]$.

Eq. 1 was corrected.

• Page 15, figure 5: especially considering the number of points, it may be good to have the corresponding data in the SI.

Having the corresponding data for every point on this figure for the various samples, concentrations, temperatures, and fields would be an endless dataset or table that would be difficult to follow. The data can be accessed at https://doi.org/10.5281/zenodo.15112854

• Page 15, figure 5: what are the corresponding β values?

For Gd-PyMTA, the values are partially shown in Fig. 9A. For Gd-TPMTA, the values are partially shown in Fig. 7F and Fig. 9B. In the temperature range from 1.6 to 10 K, the beta values are in the range of 0.6-0.9 and do do change much with the temperature.

• Page 16, lines 18-19: "while holding the time between the first π/2 pulse and the last echo constant equal to 2τ". It may be better to reformulate this as τ is not constant throughout the experiment.

This was changed to "while holding the time between the first $\pi/2$ pulse and the last echo equal to $2\tau$ for all n"

• Page 16, lines 26-27: "β exhibits a monotonic decrease from n=1 to n=3 and levels off at n≥3, where it reaches a value of 1". β for 200 μM Gd-PyMTA seems to be consistently above 1 (Fig. 7D) whereas β for Gd-TPMTA seems to reach an asymptotic value < 1 (Fig. 7E).

We changed to "β exhibits a monotonic decrease from n=1 to n=3 and levels off at n≥3, where it reaches a value of 1-1.2......................... The behavior of Gd-TPMTA is similar but less pronounced; $1/T_m$ decreases from $n$=1 to $n$=2 but then levels off and $\beta$ levels off between $n$=3 and n=4 and reached 0.9."

• Page 18, Figure 7: it may be better to report the concentration of the Gd-TPMTA sample.

Was added to the caption.

• Page 20, line 2 (caption of Figure 8): the reported parameters do not seem to be related to Equation 2 (Af + A ≠ 1, no reference to Af in Equation 2).

This was corrected.

• Page 20, line 16: "This might be why full CP train refocuses the SD contributions better". What does "This" refer to? E.g., τ = τmin in the full CP train experiment? Were full CP train traces recorded for τ values larger than τmin to check that indeed τ = τmin yields the best refocussing conditions?

We changed to : The short τ used in the full CP train seemed to refocus the SD contributions better. We verified that by increasing τ in the full CP train, the decay rate increased.

• Page 22, Figure 10: What were the used concentrations?

Was added

• Page 22, Figure 10: hy is there no datapoint for Gd-TPMTA at 20 K if this experiment has been reported in Fig. 9B (bottom right)?

Two spectrometers were used in this work, and the results are slightly different because the cooling rate of the samples in the two spectrometers is different, generating slightly different glass quality upon freezing. All Gd-PyMTA measurements were performed using the first home-built W-band pulse EPR spectrometers, and all temperature and field dependencies of Gd -TPMTA below shown in Fig. 10 were

carried out using the second, DNP spectrometer due to its wide temperature and magnetic field ranges. The datapoint for Gd-TPMTA at 20 K was not measured on this spectrometer and as we do not want to mix data that were carried out on different spectrometers, we did not use the point from Fig. 9B.

• Page 22, lines 5-6: "For the Hahn echo, we observed a clear enhancement of the decay rate outside the CT". Is it worth referring the reader to Fig. 4 C/F?

This was added.

• Page 23, Figure 11: what do the colours and the filled/empty symbols refer to?

The caption of Fig. 11 was changed and A,B,C,D markings were added to the fig.

• Page 23, Figure 11: were the pulse amplitudes adjusted for the individual field positions? (see e.g. the reported equation in Section 2.4).

Yes

• Page 25, lines 10-11: "the contribution of the protonated HEPES is small, but that of the lower amount of glycerol is significant". How was the effect of the lower amount of glycerol interpreted? Enhanced instantaneous diffusion due to a poorer quality of the glass?

Yes, we added "Enhanced instantaneous diffusion due to a poorer quality of the glass"

• Page 26, Figure 12: to enable direct comparison, it may be useful to include the reference values (unconjugated tag) to this plot.

This is done in a new Table S3 in the SI.

• Page 26, line 4 (caption of Figure 12): "Gd-PyMTA, the results for the CPMG slow component have also been added". I am afraid I can't follow what this statement refers to: CP data for Gd-TPMTA have been reported as well. Moreover, from what I understood the Meiboom-Gill variant of the CP sequence was not performed.

Thanks, the caption was a mess. It was corrected.

Supplementary information

• Page S15, Figure S9B: why were the $1/T_m$ data plot against $t\pi$? $(1/T_m)ID$ is expected to be proportional to $\sin^2(\theta 2/2)$, see Equation S1. As a related point, are the experimental results consistent with the theoretical prediction?

We added to Fig. S9 a plot of 1/Tm as a function of $\sin^2(\theta2/2)$. The data agree very well with the theoretical prediction. We added to the caption: B. The dependence of $1/T_m$ on $\sin^2(\theta_2/2)$, where $1/T_m$ is given in C and $\theta_2$ is the flip angle, with 30 ns corresponding to a $\pi$ pulse (see eq. S1). Linear regression gave a slope of $-6.9\times10^{-3}\pm4.2\times10^{-4}$ $\mu s^{-1}$ and an intercept of $0.082\pm0.0003$ $\mu s^{-1}$. The slope obtained shows a remarkable agreement with the calculated contribution for 200 $\mu$M according to eq. S1 giving a slope of $-6.94\times10^{-3}$ $\mu s^{-1}$. This confirms the very small contribution of instantaneous diffusion to the phase relaxation.

• Page S21: "As shown by Mitrikas and co-workers". Does this reference to Mitrikas 2023?

No this will be published elsewhere as noted in p. 21.

• Page S24, caption of Figure 20: it would be beneficial to specify in the caption what the sample labels (e.g., Gd-PyMTA HEPES 8:2) refer to.

Done

Technical corrections

• Page 9, line 1: "T1 measurements for were performed using the inversion recovery sequence, $\pi$ – twait – $\pi/2$ – $\tau$ – $\pi$ – $\tau$ – echo, with varying twait". Please, remove the "for".

We did not find the "for"

• Page 9, line 29: *10K → 10 K (add a space between the figure and the units). This occurs in other places throughout the text.

done

• Page 11, line 14: *phenyl rings → phenyl ring (GdPyMTA has only one phenyl ring)

done

• Page 13, line 15: where the central transition exhibits a longer Tm; a characteristic of the tZFS mechanism. It may be better to replace the semicolon with a comma.

done

• Page 14, line 7 (caption of Figure 4): *(µs,µM)-1 → (µs·µM)-1 or µs-1 /µM.

done

• Page 14, line 9 (caption of Figure 4): in the equation for the dotted lines in panels B and E the x axis is a concentration; the parameter T should have the same units.

done

• Page 15, line 6: the sentence "a linear correlation was observed only for the 5-15K;" is incomplete.

Added 5-15 K range.

• Page 21, Figure 9B, bottom right: do the numbers in brackets refer to the different deuteration conditions?

No, these are the different fields (explained in the caption)

• Page 24, line 14: *for Gd-TPMTA labelled ubiquitin, a significant effect was noticed. I would consider removing the comma.

Done

 • SI, page S14, notes to Table S1: *Gd-F distance → Gd-H distance.

Done

• SI, page S15, Figure S10: for the sake of a better presentation it may be more convenient to estimate the magnetisation at the thermal equilibrium by averaging a few points at the end of each trace; this would ensure that all the traces have 1 as their asymptotic value.

Good idea, but was not done and there is no point to repeat the measurements now.

 • SI, page S16, Figure S11: as both $T_1^{-1}$ and $T_m^{-1}$ are reported on the same plot, it may be better to use a logarithmic y scale.

Thank you, good idea, done.

 • SI, page S16, beginning of section 7: *the contributions of TID → the contribution of TID.

Done

• SI, page S20: multiple spacing in *pulses are so close.

Done

• SI, page S23, caption of Figure S19: *Af → A to keep consistency with Equation 2 in the main text.

Done

---

## Author Comment (AC3)

**Reviewer 1 – Gunnar Jeschke**

While understanding of decoherence of electron spins in nitroxide spin labels improved substantially during the past few years, data for Gd(III) spin labels were relatively scarce, in particular at high frequencies (95 GHz) where these labeles perform particularly well. The current manuscript addresses this gap in a systematic way and presents very interesting results. Experiments and data analysis are stateof-the-art, data quality is high, and the presentation is clear. I have little to criticize. The following points should be addressed in minor revision.

**Thank you !**

1. The manuscript would profit from a Table that provides an overview of Tm (or relaxation rate 1/Tm) for the various samples.

Making such a table for the various samples (labels and proteins), concentrations and temperatures and fields would be an endless table which will be hard to follow. We will present the following table for conditions that are usually used in DEER, namely 10K and 50  $\mu$ M (or 25  $\mu$ M), measured at the central transition.

Table : Overview of the  $T_m$  values of the samples studied in this work measured by Hahn at the CT and 10 K. For Gd-TMTPA the values measured at fields 1,2,and 3 are averaged.

| Sample                       | Conc. | Τ m (μs) |                        | β                      |                        |
|------------------------------|-------|----------------------------|------------------------|------------------------|------------------------|
|                              | (μM)  |                            |                        |                        |                        |
| Gd-PyMTA                     | 50    | 16.53 ± 0.77               |                        | 1.21 ± 0.07            |                        |
| Gd-PyMTA-d 8,     | 50    | 19.59 ± 0.31               |                        | 1.14 ± 0.05            |                        |
| Gd-PyMTA-d 12     | 50    | $18.41 \pm 0.84$           |                        | $1.16 \pm 0.08$        |                        |
| Gd-TPMTA                     | 50    | 9.96 ± 0.57                |                        | 0.92 ± 0.03            |                        |
| Gd-TPMTA-d 8,     | 50    | 10.09 ± 1.33               |                        | 0.94 ± 0.07            |                        |
| Gd-TPMTA-d 12     | 50    | 9.75 ± 0.42                |                        | $0.90 \pm 0.04$        |                        |
|                              |       | 1 H protein     | 2 H protein | 1 H protein | 2 H protein |
| Ubi-Gd-PyMTA                 | 25    | 8.1 ± 0.04                 | 9.73 ± 0.05            | $1.43 \pm 0.01$        | $1.38 \pm 0.01$        |
| Ubi-Gd-PyMTA-d 8, | 25    | 8.25 ± 0.04                | 9.1 ± 0.05             | 1.33 ± 0.01            | $1.27 \pm 0.01$        |
| Ubi-PyMTA-d 12    | 25    | 8.18 ± 0.03                | 9.1 ± 0.07             | 1.35 ± 0.01            | 0.99 ± 0.01            |
| Ubi-Gd-TPMTA                 | 50    | 5.4 ± 0.6                  | 5.2 ± 0.3              | $1.19 \pm 0.04$        | $0.99 \pm 0.01$        |
| Ubi-Gd-TPMTA-d 8, | 50    | 6.2 ± 0.6                  | 6.6 ± 0.4              | 1.02 ± 0.05            | $1.00 \pm 0.02$        |
| Ubi-Gd-TPMTA-d 12 | 50    | 5.6 ± 0.3                  | 9.1 ± 0.05             | 1.02 ± 0.02            | 1.27 ± 0.01            |
| Ub-Gd-DO3A                   | 50    | 8.98 ± 0.04                | 8.09 ± 0.04            | $1.05 \pm 0.01$        | 0.912 ± 0.004          |
| Ub-Gd-DO3A-d 8    | 50    | 8.77 ± 0.04                | 9.34± 0.05             | $1.014 \pm 0.004$      | 0.916 ± 0.004          |

 In the Conclusion (point 3), the authors discuss residual nuclear spin diffusion as a contribution to 1/Tm for C -> 0 and focus this discussion on only the label protons. A potential contribution from residual protons in the deuterated matrix should be mentioned.

We will change to "At the limit of  $[C] \rightarrow 0$ , the contributions to  $T_m$  (0) can be residual NSD of the protons on the pyridine rings with hyperfine couplings below 0.4 MHz or residual protons in the deuterated matrix, tZFS, and direct  $T_1$ ."

3. In principle, simulation tools exist for predicting the contribution of the label protons to 1/Tm (at least for the Hahn echo/CP1 case). While such predictions may be beyond the scope of the current manucsript, I encourage the authors to address this issue in the future, also relating this to point 2 (residual matrix protons).

We will add in point 3 of the conclusions : "In principle, it would possible to predict the contribution of the above mentioned weakly coupled protons and residual solvent protons to the Hahn echo decay using the analytical pair product approximation which allows for computationally efficient simulations and provides a good prediction.(Canarie et al., 2020; Jeschke, 2023). This, however, is beyond the scope of this manuscript."

 Given the importance of Gd(III) longitudinal relaxation as a contrubtion to 1/Tm, it would be helpful to include a paragraph with a few references to previous work on T1 of Gd(III).

**We will add the following to page 5:**

The T1 values of Gd(III) complexes in solution are relatively short and therefore expected it to affect the Gd(III) phase relaxation. For example, Gd(III) ruler with a PyMTA chelate with distances of 3.4 nm has at W-band T1 values in the range of 80-11  $\mu$ s at the temperature range of 6-30 K respectively, (Seal et al., 2022), (Razzaghi et al., 2014). For the same type of ruler with distances of 2.1 and 6 nm T1 of ~30  $\mu$ s was reported at 10 K (Mocanu et al., 2025). The reported T1 values of the spin label BrPsPy-DO3A-Gd(III) in the temperature range of 6-40 K are 132-9  $\mu$ s (Seal et al., 2022). At Q-band the T1 values are longer than at W-band; for the complexes of the [GdIII(NO3Pic)] family, which have a small ZFS with D $^{500}$  MHz T1 in the range of 190-200  $\mu$ s was reported (Ossadnik et al., 2023).

 Reference (Pannier et al., 2011) points to a 10th anniversary reprint of the original paper [ (Pannier et al., 2000, https://doi.org/10.1006/jmre.1999.1944)]. It might be more appropriate to cite the original paper

Oops, sorry about this, will be fixed.

**References:**

Canarie, E. R., Jahn, S. M., and Stoll, S.: Quantitative Structure-Based Prediction of Electron Spin Decoherence in Organic Radicals, J. Phys. Chem. Lett., 11, 3396-3400, 10.1021/acs.jpclett.0c00768, 2020.

Jeschke, G.: Nuclear pair electron spin echo envelope modulation, Journal of Magnetic Resonance Open, 14-15, 100094, https://doi.org/10.1016/j.jmro.2023.100094, 2023.

Mocanu, E. M., Ben-Ishay, Y., Topping, L., Fisher, S. R., Hunter, R. I., Su, X.-C., Butler, S. J., Smith, G. M., Goldfarb, D., and Lovett, J. E.: Robustness and Sensitivity of Gd(III)–Gd(III) Double Electron–Electron Resonance (DEER) Measurements: Comparative Study of High-Frequency EPR Spectrometer Designs and Spin Label Variants, Appl. Magn. Reson., 56, 591-611, 10.1007/s00723-024-01741-0, 2025.

Ossadnik, D., Kuzin, S., Qi, M., Yulikov, M., and Godt, A.: A GdIII-Based Spin Label at the Limits for Linewidth Reduction through Zero-Field Splitting Optimization, Inorganic Chemistry, 62, 408-432, 10.1021/acs.inorgchem.2c03531, 2023.

Razzaghi, S., Qi, M., Nalepa, A. I., Godt, A., Jeschke, G., Savitsky, A., and Yulikov, M.: RIDME Spectroscopy with Gd(III) Centers, J. Phys. Chem. Lett., 5, 3970-3975, 10.1021/jz502129t, 2014.

Seal, M., Feintuch, A., and Goldfarb, D.: The effect of spin-lattice relaxation on DEER background decay, J. Magn. Reson., 345, 107327, https://doi.org/10.1016/j.jmr.2022.107327, 2022.

---

## Author Response (AR1)

Dear Editor

Here is our response to the reviewers. Their comments were copied (black fonts), and our response, along with a description of changes made in the manuscript (blue fonts). We also made minor changes in the manuscript according to the comments of Alberto Collauto. These, along with the changes made in response to the reviewers, are highlighted in a copy of the revised manuscript.

 Reviewer 1 – Gunnar Jeschke

While understanding of decoherence of electron spins in nitroxide spin labels improved substantially during the past few years, data for Gd(III) spin labels were relatively scarce, in particular at high frequencies (95 GHz) where these labeles perform particularly well. The current manuscript addresses this gap in a systematic way and presents very interesting results. Experiments and data analysis are state-of-the-art, data quality is high, and the presentation is clear. I have little to criticize. The following points should be addressed in minor revision.

Thank you !

1.  The manuscript would profit from a Table that provides an overview of Tm (or relaxation rate 1/Tm) for the various samples.

    Making such a table for the various samples (labels and proteins), concentrations and temperatures and fields would be an endless table which will be hard to follow.  We added a table to the SI  for conditions that are usually used in DEER, namely 10K and 50 μM (or 25  μM), measured at the central transition.

**Table S3** : Overview of the $T_m$ and $\beta$ values of the samples studied in this work measured by Hahn echo at the CT and 10 K.

| Sample | Conc. (µM) | $T_m$ (µs) | | $\beta$ | |
|---|---|---|---|---|---|
| Gd-PyMTA | 50 | 16.53 ± 0.77 | | 1.21 ± 0.07 | |
| Gd-PyMTA-d$_8$, | 50 | 19.59 ± 0.31 | | 1.14 ± 0.05 | |
| Gd-PyMTA-d$_{12}$ | 50 | 18.41 ± 0.84 | | 1.16 ± 0.08 | |
| Gd-TPMTA[a] | 50 | 9.96 ± 0.57 | | 0.92 ± 0.03 | |
| Gd-TPMTA-d$_8$[a] | 50 | 10.09 ± 1.33 | | 0.94 ± 0.07 | |
| Gd-TPMTA-d$_{12}$[a] | 50 | 9.75 ± 0.42 | | 0.90 ± 0.04 | |
| | | [1]H protein | [2]H protein | [1]H protein | [2]H protein |
| Ubi-Gd-PyMTA | 25 | 8.1 ± 0.04 | 9.73 ± 0.05 | 1.43 ± 0.01 | 1.38 ± 0.01 |

| | | | | | |
|---|---|---|---|---|---|
| Ubi-Gd-PyMTA-$d_8$ | 25 | 8.25 ± 0.04 | 9.1 ± 0.05 | 1.33 ± 0.01 | 1.27 ± 0.01 |
| Ubi-PyMTA-$d_{12}$ | 25 | 8.18 ± 0.03 | 9.1 ± 0.07 | 1.35 ± 0.01 | 0.99 ± 0.01 |
| Ubi-Gd-TPMTA[a] | 50 | 5.4 ± 0.6 | 5.2 ± 0.3 | 1.19 ± 0.04 | 0.99 ± 0.01 |
| Ubi-Gd-TPMTA-$d_8$[a] | 50 | 6.2 ± 0.6 | 6.6 ± 0.4 | 1.02 ± 0.05 | 1.00 ± 0.02 |
| Ubi-Gd-TPMTA-$d_{12}$[a] | 50 | 5.6 ± 0.3 | 9.1 ± 0.05 | 1.02 ± 0.02 | 1.27 ± 0.01 |
| Ub-Gd-DO3A | 50 | 8.98 ± 0.04 | 8.09 ± 0.04 | 1.05 ± 0.01 | 0.912 ± 0.004 |
| Ub-Gd-DO3A-$d_8$ | 50 | 8.77 ± 0.04 | 9.34± 0.05 | 1.014 ± 0.004 | 0.916 ± 0.004 |

[a]Average of measurements carried out at fields 1,2, and 3.

2. In the Conclusion (point 3), the authors discuss residual nuclear spin diffusion as a contribution to 1/Tm for C -> 0 and focus this discussion on only the label protons. A potential contribution from residual protons in the deuterated matrix should be mentioned.

   We changed to "At the limit of [C]→0, the contributions to $T_m$ (0) can be residual NSD of the protons on the pyridine rings with hyperfine couplings below 0.4 MHz or residual protons in the deuterated matrix,  tZFS, and direct $T_1$."

3. In principle, simulation tools exist for predicting the contribution of the label protons to 1/Tm (at least for the Hahn echo/CP1 case). While such predictions may be beyond the scope of the current manucsript, I encourage the authors to address this issue in the future, also relating this to point 2 (residual matrix protons).

   We added in point 3 of the conclusions : "In principle, it would possible to predict the contribution of the above mentioned  weakly coupled protons and residual solvent protons to the Hahn echo decay using the analytical pair product approximation which allows for computationally efficient simulations and provides a good prediction.(Canarie et al., 2020; Jeschke, 2023). This, however, is beyond the scope of this manuscript."

4. Given the importance of Gd(III) longitudinal relaxation as a contrubtion to 1/Tm, it would be helpful to include a paragraph with a few references to previous work on T1 of Gd(III).

We added the following to page 5:

The $T_1$ values of Gd(III) complexes in solution are relatively short and therefore expected it to affect the Gd(III) phase relaxation.  For example, Gd(III) ruler with a PyMTA chelate with distances of 3.4 nm has at W-band $T_1$ values in the range of 80-11 $\mu$s at the temperature range of 6-30 K respectively, (Seal et al., 2022), (Razzaghi et al., 2014). For the  same type of ruler with distances of 2.1 and 6 nm $T_1$ of ~30 $\mu$s was reported at 10 K (Mocanu et al., 2025). The reported $T_1$ values of the spin label BrPsPy-DO3A-Gd(III) in the temperature range of 6-40 K are 132-9 $\mu$s (Seal et al., 2022). At Q-band the $T_1$ values are longer than at W-band; for the complexes of the [Gd$^{III}$(NO3Pic)]  family, which have a small ZFS with D~500 MHz $T_1$ in the range of 190-200 $\mu$s was reported (Ossadnik et al., 2023).

5. Reference  (Pannier et al., 2011)  points to a 10th anniversary reprint of the original paper [ (Pannier et al., 2000, https://doi.org/10.1006/jmre.1999.1944)]. It might be more appropriate to cite the original paper

Oops, sorry about this, was fixed.

References:

Canarie, E. R., Jahn, S. M., and Stoll, S.: Quantitative Structure-Based Prediction of Electron Spin Decoherence in Organic Radicals, J. Phys. Chem. Lett., 11, 3396-3400, 10.1021/acs.jpclett.0c00768, 2020.

Jeschke, G.: Nuclear pair electron spin echo envelope modulation, Journal of Magnetic Resonance Open, 14-15, 100094, https://doi.org/10.1016/j.jmro.2023.100094, 2023.

Mocanu, E. M., Ben-Ishay, Y., Topping, L., Fisher, S. R., Hunter, R. I., Su, X.-C., Butler, S. J., Smith, G. M., Goldfarb, D., and Lovett, J. E.: Robustness and Sensitivity of Gd(III)–Gd(III) Double Electron–Electron Resonance (DEER) Measurements: Comparative Study of High-Frequency EPR Spectrometer Designs and Spin Label Variants, Appl. Magn. Reson., 56, 591-611, 10.1007/s00723-024-01741-0, 2025.

Ossadnik, D., Kuzin, S., Qi, M., Yulikov, M., and Godt, A.: A GdIII-Based Spin Label at the Limits for Linewidth Reduction through Zero-Field Splitting Optimization, Inorganic Chemistry, 62, 408-432, 10.1021/acs.inorgchem.2c03531, 2023.

Razzaghi, S., Qi, M., Nalepa, A. I., Godt, A., Jeschke, G., Savitsky, A., and Yulikov, M.: RIDME Spectroscopy with Gd(III) Centers, J. Phys. Chem. Lett., 5, 3970-3975, 10.1021/jz502129t, 2014.

Seal, M., Feintuch, A., and Goldfarb, D.: The effect of spin-lattice relaxation on DEER background decay, J. Magn. Reson., 345, 107327, https://doi.org/10.1016/j.jmr.2022.107327, 2022.

**Reviewer 2 - Giuseppe Sicoli**

The manuscript represents an interesting example of understanding to design and design for understanding related to the impact of the deuteration within the context of the relaxation processes. The message of the manuscript is clear and exhaustively delivered; however, for fulfilling the main 'take-home' message of the manuscript, some minor points can be revised:

Thank you.

- For the comparison of the values of $1/T_{m,f}$ and $b_f$ for different samples and temperatures (figure 7 C, F; please notice into the text at page 20 such figure has been referred as '5'), the authors refer to Figure 10 (page 22) for describing the contribution of the fast component. Besides the *fairly constant behaviour* for the PyMTA, it would be interesting to provide further elements to the discussion on the behaviour of TPMTA, exhibiting a completely different behaviour.

  Thank you; we changed 5 to 7.

  As for the different behavior of TPMTA, we added in p. 21 the following :

  The relative contribution of the two components is fairly constant in the temperature range tested for Gd-PyMTA, whereas for Gd-TPMTA the contribution of the fast component is constant for 1.6-4 K and thereafter, a significant increase with increasing temperature is observed in the range of 6-15 K **(Fig. 10)**. This trend seems to correlate with the relative intensity of the central transition (**Fig. 2**). Currently we do not have an explanation for this behavior.

- The general approach proposed does not mention the effect of the pH, which may have an impact into the affinity of the two main ligands described; such an effect on the relaxation is probably beyond the scope of the manuscript, but it can be worth to mention also that tuneable parameter (i.e., pH).

  We usually prepare the spin label at a pH where all carboxylates are deprotonated and able to coordinate the Gd(III), changing pH may lead to variation in the number of ligands and the ZFS, and will complicate things. We therefore think that this should be a completely different study.

- The assignment of dominating mechanism assigned for the two populations (*slow* and *fast*), as summarized on page 23 (lines 8-11) can eventually be reinforced by citing known structures where the $T_1$ and tZFS are distinctively contributing to the relaxation paths. It may support the effect of the deuteration for 'small' molecules and validate the less pronounced effect on labelled proteins.

  Unfortunately, as far as we know, there are no such studies, except the study of Raitsimring. It would be nice to have a correlation of $T_1$ with the ZFS, but currently, there is not enough data (measured at the same frequency and temperature) to support such a correlation.

- Please notice that the authors refer to Figure 2D (page 10) but the capital letter on the figure 2 (page 11) is missing. A-B-C-D on the four panel must be revised.

  This was fixed.